# Folding pathway of a discontinuous two-domain protein

Ganesh Agam [1,2,3], Anders Barth [1,2,4] & Don C. Lamb [1,2] ✉

It is estimated that two-thirds of all proteins in higher organisms are composed of multiple domains, many of them containing discontinuous folds. However, to date, most in vitro protein folding studies have focused on small, single-domain proteins. As a model system for a two-domain discontinuous protein, we study the unfolding/refolding of a slow-folding double mutant of the maltose binding protein (DM-MBP) using single-molecule two- and three-color Förster Resonance Energy Transfer experiments. We observe a dynamic folding intermediate population in the N-terminal domain (NTD), C-terminal domain (CTD), and at the domain interface. The dynamic intermediate fluctuates rapidly between unfolded states and compact states, which have a similar FRET efficiency to the folded conformation. Our data reveals that the delayed folding of the NTD in DM-MBP is imposed by an entropic barrier with subsequent folding of the highly dynamic CTD. Notably, accelerated DM-MBP folding is routed through the same dynamic intermediate within the cavity of the GroEL/ES chaperone system, suggesting that the chaperonin limits the conformational space to overcome the entropic folding barrier. Our study highlights the subtle tuning and co-dependency in the folding of a discontinuous multi-domain protein.

The efficient folding of proteins into their three-dimensional functional structure is of fundamental importance for cell viability. In vitro studies of protein folding have provided detailed insights into the folding process by offering precise control of experimental conditions[1,2]. To date, most folding studies have focused on simple, small proteins of up to ~100 amino acids, while little is known about the folding of multi-domain proteins that comprise more than two-thirds of the eukaryotic proteome[3–7]. Notably, ~28% of all multidomain proteins are discontinuous[8], which lowers the efficiency of the folding process as the domains are co-dependent but allows for more functionality such as allosteric coupling between the domains[9–11]. Compared to the fast, spontaneous folding of small proteins (microseconds to milliseconds)[12], large multidomain proteins fold relatively slowly (seconds to minutes) and exhibit a complex multi-phase folding process[6,13]. The co-dependent folding of multi-domain proteins needs

a fine tuning of the folding of the individual domains to allow the correct folding order or pathway as suggested by theoretical studies[14,15]. Nature tackles the challenge of folding these complex topologies on physiological time-scales by sculpting of the folding energy landscape either within the structure, by using co-translational folding on the ribosome or with the help of chaperones[16–19].

Here, we investigate the folding of the maltose binding protein (MBP), a monomeric ~42 kDa two-domain protein comprised of a discontinuous N-terminal domain (NTD) and a C-terminal domain (CTD) (Fig. 1a). The double mutation V8G/Y283D in the NTD[20] has been shown to significantly slow down the folding time (DM-MBP, half-life $t_{1/2}$ ~ 30 min) compared to wildtype MBP (WT-MBP, $t_{1/2}$ ~ 25 s) and can be rescued by the chaperonin GroEL/ES system[21]. It has been speculated that the two mutations present in DM-MBP delay the formation of a nucleation core in the NTD by weakening the hydrophobic

[1]Department of Chemistry, Ludwig-Maximilians University Munich, Munich, Germany. [2]Center for NanoScience, Ludwig-Maximilians University Munich, Munich, Germany. [3]Present address: MRC Laboratory of Molecular Biology, Francis Crick Avenue, Cambridge Biomedical Campus, Cambridge CB2 0QH, UK. [4]Present address: Department of Bionanoscience, Kavli Institute of Nanoscience Delft, Delft University of Technology, 2629HZ Delft, The Netherlands. ✉e-mail: d.lamb@lmu.de

interactions (V8G, Y283D)[22]. The folding order has been observed for WT-MBP in a hydrogen exchange-mass spectroscopy (HX-MS) study where the NTD folded over an order of magnitude faster ($t_{1/2} \sim 1\,s$) compared to the CTD ($t_{1/2} \sim 40\,s$)[23]. Interestingly, DM-MBP exhibited intermediate states of unknown structure during the folding process in ensemble assays[24,25]. We investigate the unknown nature of the folding intermediate and whether the order of domain folding of DM-MBP is preserved with respect to the WT using two- and three-color single-molecule Förster Resonance Energy Transfer (smFRET) experiments. Our results provide experimental evidence for the cooperative folding within multi-domain proteins by assessing the role of entropy. The intermediate population was found to be dynamic, fluctuating between the unfolded state and a compact state or states from which the protein can fold into the native configuration. We could also establish that the folding hierarchy of the two domains is maintained. Hence, the entropic search controls the folding of the NTD and thus the entire folding rate. Lastly, we show the existence of this dynamics between unfolding and compact states within the chaperonin but with a shift to compact conformations, leading to the efficient folding of DM-MBP.

## Results

### DM-MBP populates a unique intermediate during refolding

First, we monitored the unfolding and refolding of MBP at the ensemble level using tryptophan fluorescence (Supplementary Note 1, Fig. 1b, Fig. S1a). Samples were allowed to equilibrate under unfolding/refolding conditions for 20 hours before measuring. The unfolding/refolding curves reveal a hysteresis for DM-MBP that is absent for the wildtype protein[26], in agreement with a previous report[25] (Fig. S1a, b, Table S1). We confirmed that the delayed refolding is not caused by aggregation using fluorescence cross-correlation spectroscopy (FCCS) experiments during refolding of DM-MBP. Equal amounts of a single-cysteine mutant of DM-MBP (A52C) labeled with either Atto532 or Alexa647 were denatured in 3 M GuHCl at a concentration of 500 nM. The samples where then mixed and diluted by 25-fold to a final total protein concentration of 40 nM in 0.1 M GuHCl. The final denaturant concentration of 0.1 M GuHCl was chosen as refolding to the native state still occurs under these conditions. There was no detectable cross-correlation amplitude during 60 minutes of refolding, suggesting that the delayed refolding of DM-MBP is caused by a slowly refolding population (Supplementary Note 1, Fig. S1c).

Next, we monitored the unfolding and refolding of MBP with smFRET using pulsed interleaved excitation and multi-parameter fluorescence detection (MFD-PIE)[27]. Double-cysteine mutants of DM-MBP were labeled with the fluorophores Atto532 (donor) and Alexa647 (acceptor). The smFRET measurements exhibited a FRET efficiency of ~0.85 for the native NTD (monitored between positions 52 and 298; Fig. 1a,c, first row of FRET efficiency histograms and Fig. S2), corresponding to an interdye distance of 48 Å, which is consistent with the structure of the folded state (Table S2, S3, Fig. S3a)[28,29]. We measured the unfolding/refolding trajectory of the NTD conformation at different GuHCl concentrations after allowing the sample to equilibrate for 2 hours (Fig. 1b, c). The FRET efficiency of the unfolded state decreased from ~0.3 to ~0.1 at high GuHCl concentrations ($\geq 1\,M$) due to further expansion of the denatured polypeptide chain. At 2 M GuHCl, the NTD is fully unfolded as expected with a FRET efficiency of ~0.1 (Table S3; Fig. 1c, last row of FRET efficiency histograms). In addition to the native and unfolded state, the single-molecule FRET efficiency histograms reveal an additional population with a FRET efficiency of ~0.6 that is visible both during unfolding and refolding (Fig. 1c, orange- and yellow-dashed lines). Notably, this intermediate population remained highly populated during refolding even at low denaturant concentrations below 0.5 M, which coincides with the hysteresis range in the ensemble experiments (Fig. 1b). We note that there are differences between the unfolding/refolding curves measured via tryptophan

fluorescence versus FRET efficiency. While both methods are used to investigate protein unfolding/refolding, tryptophan fluorescence measures the solvent accessibility due to quenching by water, whereas FRET measures the actual distance between the fluorophores. Hence, some differences are expected. Next, we measured the refolding kinetics by quick dilution of unfolded DM-MBP to decrease the denaturant concentration from 3 M to below 0.1 M (Fig. S3b-c, Table S1). The NTD shows an initial FRET efficiency of 0.6, similar to the intermediate population observed under equilibrium conditions, and completely folds to the native state with a 0.85 FRET efficiency with a half-life of $29.8 \pm 0.4$ min. This rate is consistent with the refolding of the unlabeled protein, verifying that the fluorophores do not interfere with the refolding process of DM-MBP (Table S1).

We investigated whether the observed intermediate population is also present in the folding of the CTD and the formation of the N-C interface using the previously characterized double-cysteine mutants of DM-MBP; 175C-298C and 52C-175C (Fig. 2a, Fig. S3a)[24]. Indeed, smFRET measurements on the CTD and N-C interface (Supplementary Note 2, Fig. 2a, Fig. S3b-h, Fig. S4) show a similar trend as for the NTD (Supplementary Note 2, Table S1) with both a visible refolding hysteresis and an intermediate population. The NTD and N-C interface show a significantly higher fraction of folded conformations at 0.1 M GuHCl compared to the CTD, suggesting that the folding of the CTD is more impaired by the intermediate population (grey line in Fig. 2a). Notably, this intermediate population is not observed in the unfolding and refolding curves of the wildtype protein (Fig. 2c, Fig. S5).

### The intermediate population is dynamic

To investigate the nature of the intermediate population, we used the fluorescence lifetime information available in the smFRET experiments with MFD-PIE (Fig. S2). A plot of the intensity-based FRET efficiency versus the donor fluorescence lifetime ($E$-$\tau$ plot) indicates whether the molecule is static or dynamic on the ns-ms timescale. Dynamic samples will deviate from the static FRET-line (black line, Fig. 2b)[30]. For all three constructs, the intermediate population is shifted from the static FRET-line, indicating the presence of conformational dynamics (Fig. 2b, Table S4). The observed deviation can be described by a single dynamic-FRET line (red line, Fig. 2b) connecting the unfolded and compact or native states. This indicates that the intermediate population is not a stable structural state but originates from dynamic switching between the unfolded and compact states during the observation time (~ms). We estimated the FRET efficiencies of the interconverting states from the donor fluorescence decay (Fig. S6, Table S4), which agreed well with the FRET efficiencies of the folded and unfolded states (white boxes in Fig. 2a). The observed dynamics are not an artifact of the fluorophores as confirmed by experiments using a different acceptor dye (Fig. S7).

To investigate whether the hysteresis is due to not allowing the system enough time to reach equilibrium, we measured refolding of DM-MBP at 0.3 M GuHCl over three days. Indeed, the number of refolded proteins increased during the first 18 hours as seen from the rise in the number of molecules with a FRET efficiency of 0.85 (Fig. S8). Afterwards, no significant increase in the number of refolded molecules is observed, but there are changes in the dynamic properties of the unfolded intermediate. As no aggregation is visible after 48 hours, we attribute this behavior to nonreversible misfolding of DM-MBP (Fig. S8). This suggests that hysteresis comes from the presence of an additional kinetic pathway leading to irreversible protein denaturation on the day timescale.

As has been suggested earlier, the slow refolding of DM-MBP is thought to be due to the presence of an entropic barrier[25] meaning that, in the absence of the hydrophobic core, a wide conformational search on a flat energy landscape needs to be performed. If this were the case, we should expect a slight increase in denaturant in the low concentration regime (0.1 to 0.3 M GuHCl) to have a significant impact

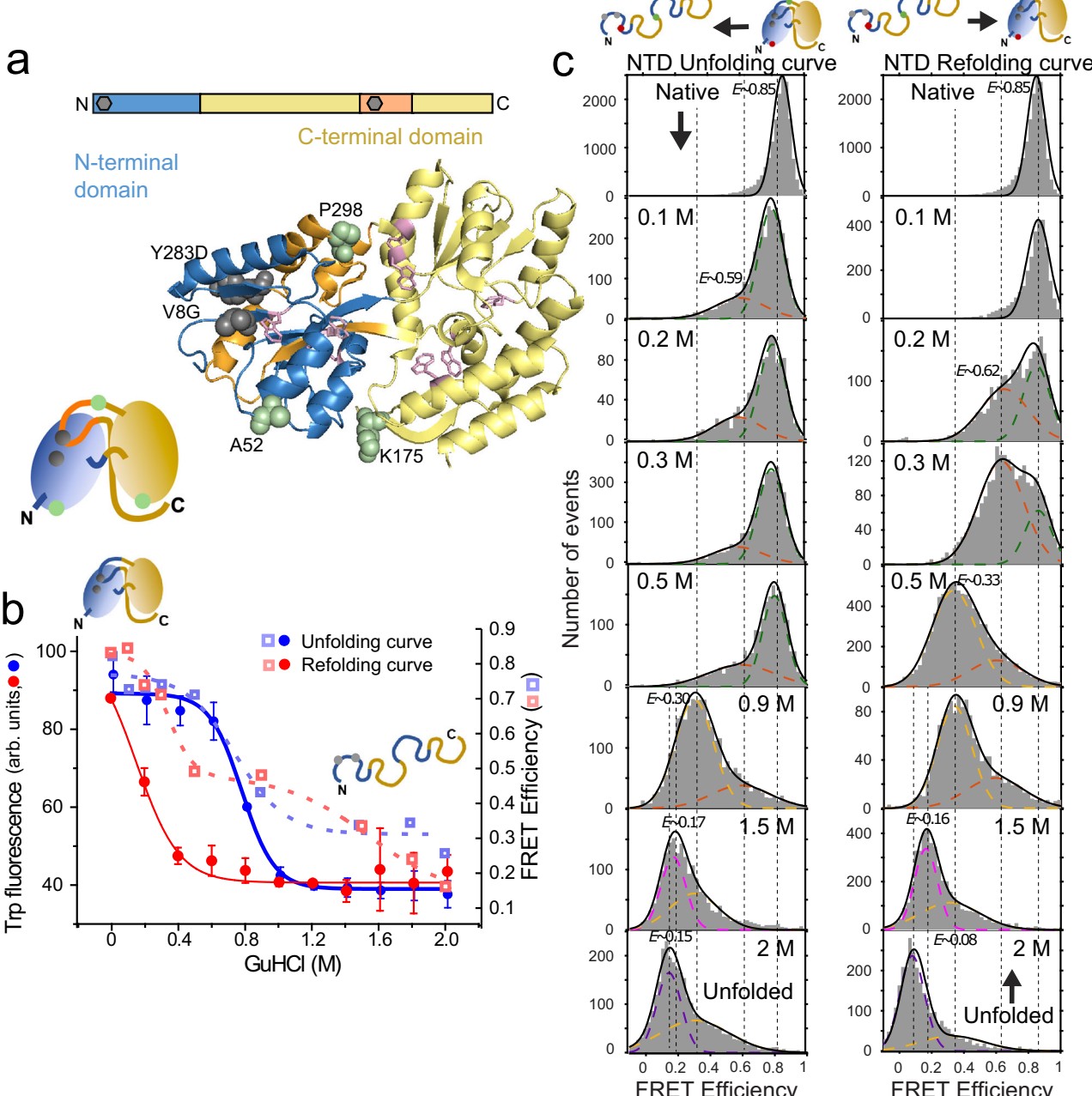

**Fig. 1 | Equilibrium unfolding-refolding two-color smFRET measurements on the DM-MBP NTD. a** A ribbon structure of MBP (PDB ID:1OMP) showing the NTD and CTD in blue and yellow, respectively with the discontinuous portion of the NTD being highlighted in orange. The upper schematic represents the domain boundaries and discontinuity in the MBP sequence for the NTD and CTD. The positions of the mutations V8G and Y283D are depicted as grey hexagons. The two residues of the folding mutations (V8G and Y283D, highlighted in dark grey) as well as the three labeling positions A52, K175, P298 (highlighted in green) for coupling the fluorescent dyes are indicated via a space-filling model. Tryptophan side-chains are highlighted using stick models in pink. **b** Protein unfolding/refolding measurements as a function of GuHCl concentration are shown for ensemble tryptophan fluorescence measurements on DM-MBP as well as for smFRET measurements for DM-MBP NTD from panel **c**. The tryptophan curves (circles) and average FRET efficiencies (pale squares) for unfolding and refolding titrations are depicted in blue and red, respectively. The lines are fits to a single or a double Boltzmann function with a single function being used for the tryptophan experiments and smFRET efficiency unfolding measurements, and a double Boltzmann function is used for the smFRET refolding measurements. For measurements of the tryptophan fluorescence, the sample was first equilibrated for 20 hours. For the single-molecule experiments, the sample was allowed to equilibrate for 2 hours (after which, no change in the smFRET histograms were observed) and then measured for 2–6 hours. Data are presented as mean values +/– SD derived from at least three independent experiments. **c** SmFRET histograms and Gaussian fits for DM-MBP NTD unfolding (left) and refolding (right) titrations at the indicated denaturant concentrations. Each underlying population is highlighted with a dotted line. The native or refolded state is shown in green, intermediate populations are shown in orange and in yellow, and the completely unfolded state is shown in magenta (1.5 M GuHCl) and violet (2 M GuHCl).

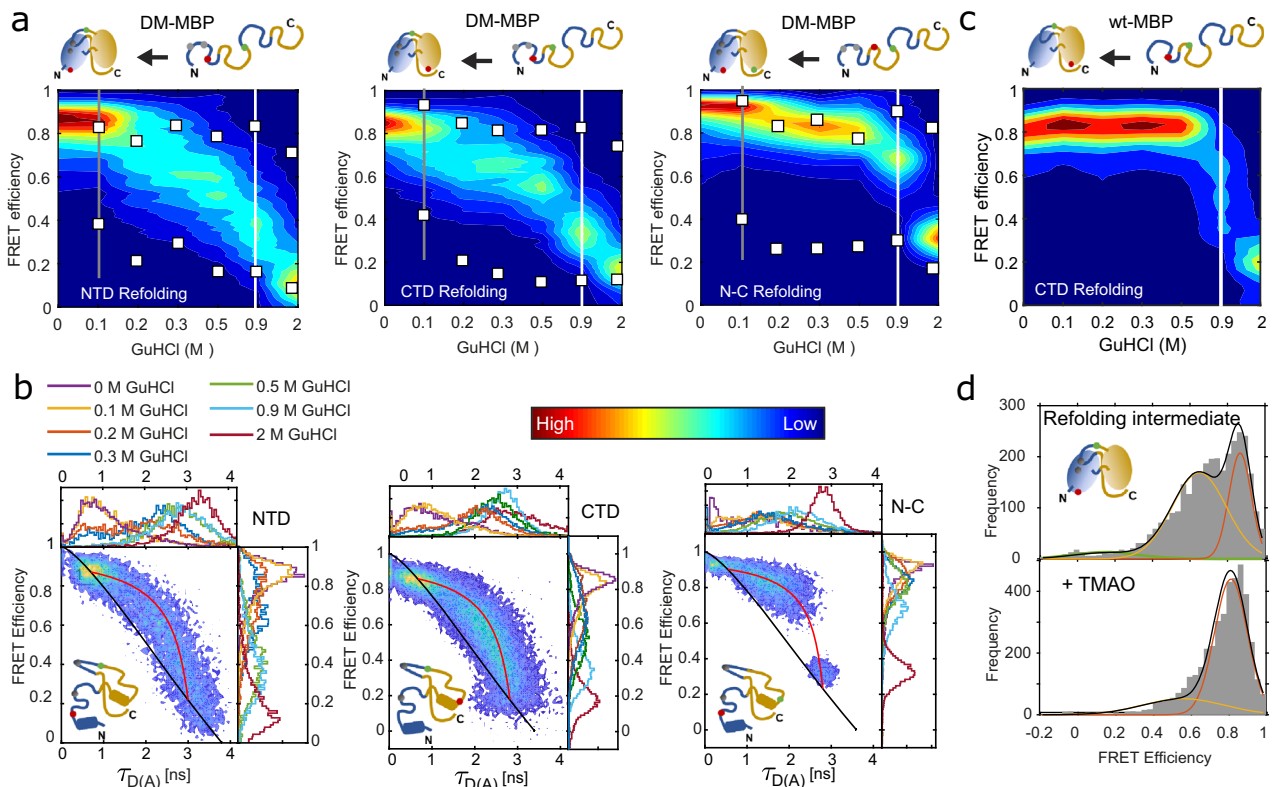

**Fig. 2 | A conformational search during DM-MBP refolding is the cause for the entropic barrier governing unfolding/refolding hysteresis. a** Equilibrium refolding of the DM-MBP NTD, CTD and N-C interface displayed as waterfall plots of FRET Efficiency versus GuHCl concentration. The burst-averaged FRET efficiencies were compared with the FRET efficiencies obtained from the two donor lifetime components determined from a fit to the photons detected from all bursts (white squares) (Table S4). A grey line is shown at 0.1 M GuHCl concentration where the NTD and N-C interface constructs exhibit a mostly refolded conformation, whereas a substantial fraction of molecules for the CTD construct are still in the inter-mediate population. A white line separates the 0.9 M GuHCl measurement from the higher denaturant concentrations. **b** 2D-plots of FRET efficiency vs donor lifetime in the presence of an acceptor ($\tau_{D(a)}$) (*E-τ* plot) is shown for the DM-MBP NTD, CTD and N-C experiments at all GuHCl concentrations displayed in panel A. The static-FRET line and a dynamic-FRET line for the measurement at 0.9 M GuHCl are shown in black and red, respectively (See Supplementary Note 2 for details). The color

legend for occurrences for both panels A and B is shown in between the two panels. **c** A waterfall plot of FRET efficiency versus GuHCl concentration is shown for equilibrium refolding measurements of the CTD of WT-MBP. A white line separates the measurements below and above 0.9 M GuHCl concentration, a concentration below which the dynamic intermediate population is significantly populated during DM-MBP refolding. Contrary to the refolding traces of the NTD, CTD, and the N-C interface for DM-MBP, an intermediate population is not clearly visible for WT-MBP refolding. **d** The influence of the chemical chaperone trimethylamine-N-oxide (TMAO). Upper panel: A smFRET histogram for DM-MBP NTD refolding in 0.2 M GuHCl, where a significant fraction of the intermediate population (yellow) is observed compared to the unfolded (green) and refolded states (orange). Lower panel: A smFRET histogram for NTD refolding in 0.2 M GuHCl where 500 mM TMAO has been added. The unfolded state is no longer visible and the intermediate population has diminished (yellow vs orange).

on the refolding rate. Denaturants are known to slow refolding and GuHCl can replace interprotein hydrogen bonds, leading to more flexibility and competition for hydrogen bonding. However, for an enthalpic barrier, it is unlikely that the small change in denaturant from 0.1 to 0.3 M would lead to more than an order of magnitude difference in refolding time. To look for additional support for an entropic barrier to refolding, we also investigated the influence of the chemical-chaperone trimethylamine N-oxide (TMAO) on DM-MBP refolding. TMAO stabilizes the solvent shell around the protein thereby confining the configuration space available to the protein[31]. DM-MBP refolding in the presence of TMAO is accelerated forming native-like structures already at 0.2 M GuHCl, where DM-MBP has a significant population of dynamic molecules in the absence of TMAO (Fig. 2d). Also, the rate of DM-MBP refolding only depends weakly on temperature (Fig. S9, Table S1 and ref. 32). Taken together, these results suggest that the refolding of DM-MBP is limited by an entropic barrier.

Next, we quantified the timescale of the dynamics observed. The *E-τ* plots indicate that conformational dynamics are faster than the burst duration of 2-3 ms (Fig. 2b). Using a species FCS analysis, which provides a model-independent assessment of the dynamic timescales,

we observed two relaxation processes in all three constructs between 0.2 and 0.9 M GuHCl concentrations: one on the fast μs timescale (4-10 μs) and a second, slower process on the 180 − 350 μs timescale (Fig. 3a, Table S5, see Fig. S11 and Supplementary Note 4 for details). We attribute the fast dynamics to unfolded polypeptide chain rearrangement[33] and the slower dynamics to large conformational changes between unfolded states and collapsed compact (hereafter compact) conformations. Interestingly, the chain dynamics are much slower than observed for small, single-domain (IDPs) proteins[34]. To quantify the microscopic rate constants of the slower conformational fluctuations, we used the dynamic photon distribution analysis (PDA) approach[35] (Supplementary Note 5, Table S6, Fig. S10a-c). The overall timescales of the relaxation rates (the inverse of the sum of the unfolding and refolding rates) determined from dynamic PDA analysis are consistent with the 180−350 μs (2-5 ms⁻¹) dynamics observed with species FCS for the same GuHCl concentrations (Fig. 3b). The apparent unfolding rates for both domains and the N-C interface increased from 0.1–1 ms⁻¹ to 2–4 ms⁻¹ with increasing denaturant concentrations. The apparent refolding rate shows a more intricate behavior (see Supplementary Note 5 for details). The CTD dynamics is roughly independent

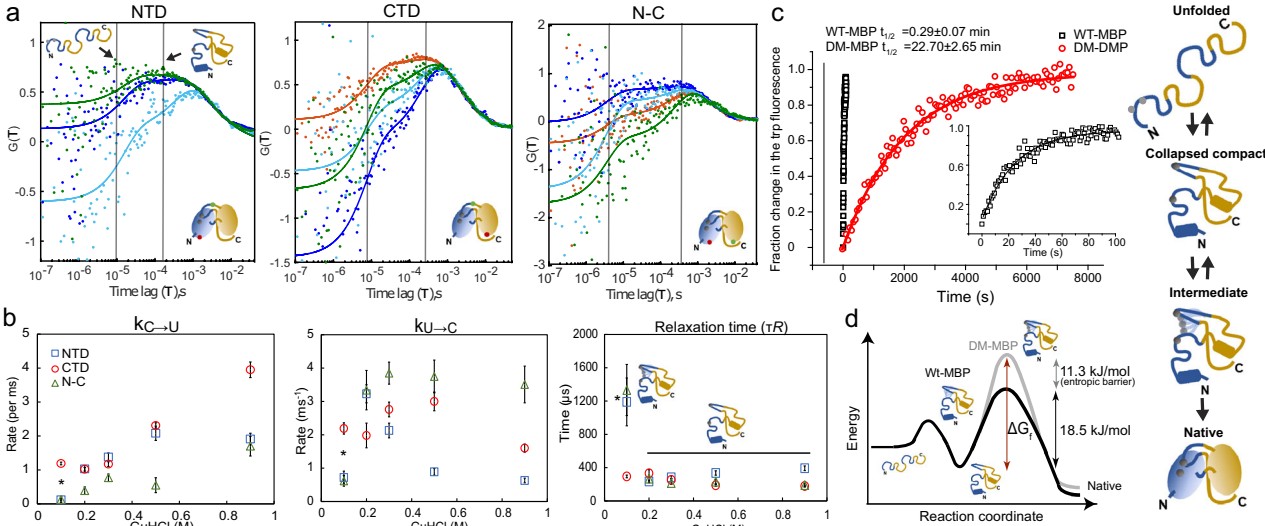

**Fig. 3 | Quantification of the entropic energy barrier. a** A cross-correlation analysis using filtered FCS on all the three DM-MBP constructs of NTD, CTD and N-C interface measured in 0.2 M (red), 0.3 M (blue), 0.5 M (green) and 0.9 M (cyan) GuHCl concentrations was performed. For clarity, only one of the two cross-correlation functions (CCFs) is shown. The CCFs at different GuHCl concentrations were fit globally for each construct (Table S5). Due to the fast refolding of the NTD at 0.2 M GuHCl, fluctuations are minimal and the FCS data have been omitted in the global fits for this construct. **b** Interconversion rates between the unfolded and collapsed compact conformations extracted from a dynamic PDA analysis during NTD refolding (blue squares), CTD refolding (red circle) and for the formation of the N-C interface (green triangle). The apparent refolding rate ($k_{C \to U}$, left), apparent unfolding rate ($k_{U \to C}$, middle) and the relaxation time (right) are plotted for all the three constructs of DM-MBP measured in 0.1, 0.2, 0.3, 0.5, 0.9 M GuHCl. The rates measured in 0.1 M GuHCl (*) were used to calculate the free energy change in

panel **d**. The obtained rates are the mean values of the fit to the model function and the errors give the 95% confidence intervals determined from the fit. **c** The kinetics of WT-MBP (black squares) and DM-MBP (red circles) refolding monitored by the increase in tryptophan fluorescence. The initial fluorescence at time t = 0 was subtracted from the subsequent data points. 3 μM MBP was denatured in 3 M GuHCl for 1 h at 50 °C before being diluted 75-fold in buffer A to start the refolding reaction (at t = 0, the final concentrations were ~40 nM of MBP and 40 mM of GuHCl). Data were fitted using a single exponential function. The fit to the WT-MBP refolding kinetics is shown in the inset for the clarity. The presented data is from a single measurement representative of three independent measurements. The given rates are the mean +/- SD from the repeats. **d** A schematic showing the Gibbs free energy versus reaction coordinate where an additional energy barrier of 11.3 kJ/mol is imposed for the refolding of DM-MBP.

of GuHCl concentration, whereas the dynamics of the NTD and N-C interface slows down at 0.1 M GuHCl where secondary and tertiary interactions are competing with the denaturant (Fig. 3b). As these interactions can be either native or non-native, the entire dynamics slows down as the protein searches for the correct conformation limiting the final folding step. In the presence of TMAO, the apparent folding rate for the NTD remains similar at 0.2 M GuHCl (2.5 ms⁻¹), whereas the apparent unfolding rate decreases significantly (0.43 ms⁻¹) (Fig. S10e, Table S6). This is the same trend observed for measurements in 0.1 M GuHCl concentration in the absence of TMAO and implies that the TMAO stabilizes the collapsed compact state at higher GuHCl concentrations. We also investigated the dynamics of the CTD during WT-MBP refolding. As WT-MBP refolds within a minute below 0.6 M GuHCl, we measured the CTD transition rates in 1 M GuHCl (Fig. S10d, Table S6). We found similar transition rates as for DM-MBP CTD, indicating a highly dynamic CTD for both WT- and DM-MBP.

Using the kinetic information, we estimated the entropic barrier induced by the folding mutations. From Kramer's equation, the difference in Gibbs free energy ($\Delta G_f$) required to cross the folding barrier is given by[2,36].

$$\tau_f \approx 2\pi\tau_0 \exp\left(\frac{\Delta G_f}{RT}\right), \qquad (1)$$

where $\tau_f$ is the folding time ($0.028\,\text{min}^{-1} = 4.7 \times 10^{-4}\,\text{s}^{-1}$), $R$ is the gas constant and $T$ is the temperature. $\tau_0$ is the reconfiguration time in the unfolded state and defines the pre-exponential factor. We estimate $\tau_0$ to be ≈1.2 ms based on the relaxation times determined from the dynamic PDA for NTD and N-C interface in 0.1 M GuHCl (Supplementary Note 5, Fig. 3b, Table S6). This yields a $\Delta G_f$ of 29.8 kJ/mol for

refolding of DM-MBP. For WT-MBP with a folding time of 0.29 min ($\tau_f = 0.057\,s^{-1}$), a $\Delta G_f$ of 18.5 kJ/mol is observed with a 11.3 kJ/mol Gibbs free energy difference to that of DM-MBP refolding (Fig. 3c).

## Three-color FRET reveals a sequential, domain-wise refolding

To directly assess the folding order of the domains, we measured the refolding of DM-MBP using three-color smFRET[37–45]. For this, it is necessary to specifically label the protein with three dyes (Supplementary Note 6, Fig. S12). Positions 52, 175 and 298 were labeled with Atto488, Alexa647 and Atto565, respectively to probe the conformation of the NTD, CTD and N-C interface via the FRET efficiencies for $E_{BG}$, $E_{GR}$ and $E_{BR}$ between the blue (b), green (G) and red (R) dyes, respectively (Fig. 4a, Fig. S13a, Table S2). The triple-labeled DM-MBP folded at a comparable rate to the unlabeled protein (Fig. S13b-f, Table S1 and Table S8), was functional (Fig. S14) and exhibited smFRET histograms similar to those from the two-color experiments (Fig. S15).

The three-color FRET unfolding and refolding experiments for DM-MBP are shown in Fig. 4b, c. To probe the folding order of the domains, we selected molecules based on their conformation in one domain and examined the conformation of the other domain (Fig. 4d). Molecules found in the refolded conformation of the NTD ($E_{BG}$) between 0.1 and 0.5 M GuHCl also exhibited a native-like N-C interface conformation whereas the CTD still has a visible intermediate component (Fig. 4d). On the contrary, most molecules that showed an intermediate population in the NTD had also exhibited an intermediate CTD ($E_{GR}$) and N-C interface ($E_{BR}$) conformation (Fig. S16). When performing a similar correlation analysis for the refolded and intermediate populations of the N-C interface, we found refolded and intermediate populations for the NTD, respectively, whereas the CTD also exhibited a significant fraction of the intermediate conformation

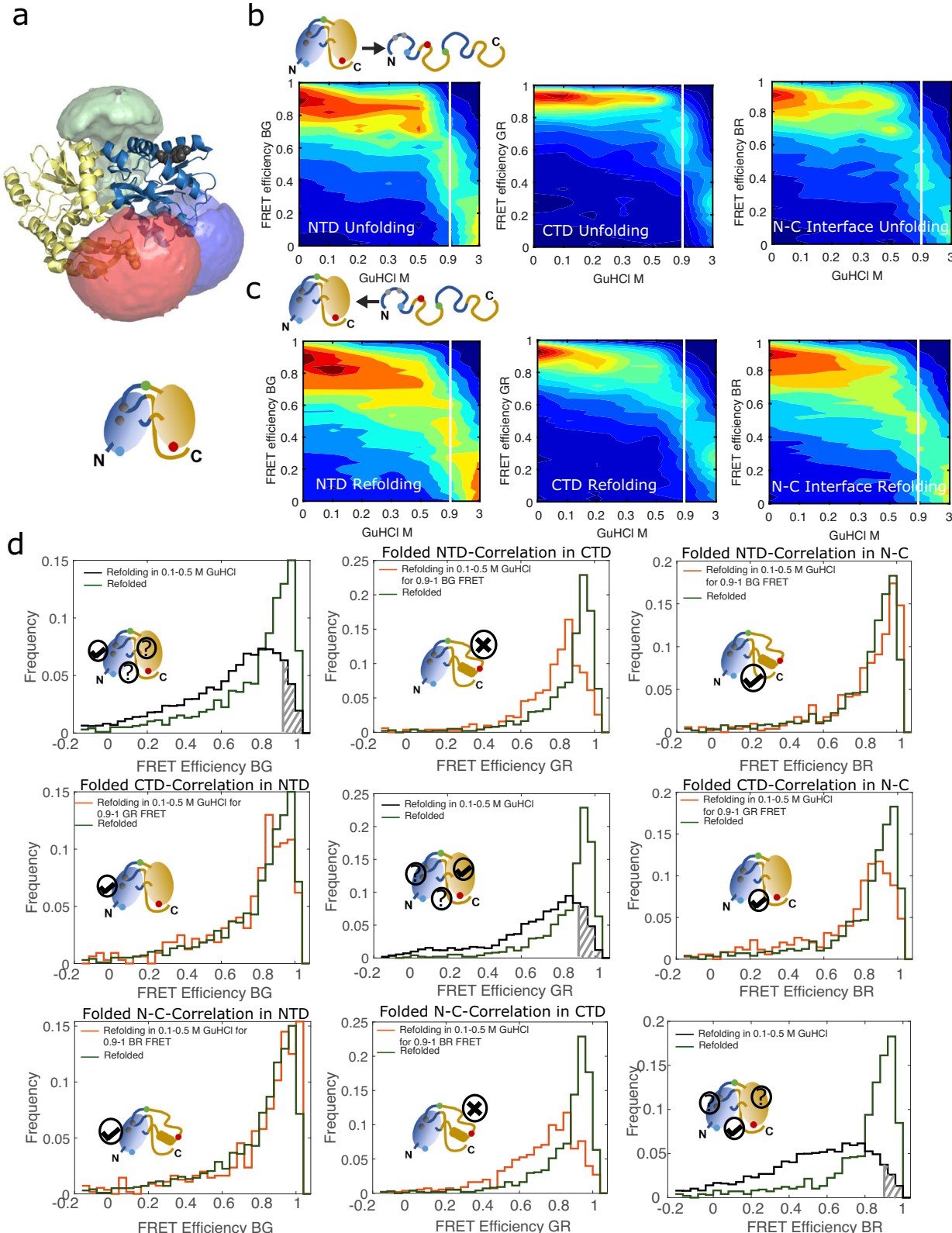

when the N-C interface has formed (Fig. 4d, Fig. S16). When analyzing molecules where the CTD is in the intermediate conformation, the intermediate population was observed for both the NTD and N-C interface. Interestingly, when the CTD is in a native-like conformation, the majority of both the NTD and the N-C interface showed a native-like conformation. There is a minor population in the intermediate conformation, which comes from non-folded CTDs being selected as

folded due to the dynamic nature of the unfolded state (Fig. 4d, Fig. S16). Taken together, these results indicate that the NTD and N-C interface fold first followed by folding of the CTD. In addition, the N-C interface can form when the α-helices harboring the 175 labeling position in the CTD is stabilized together with the folded NTD. This is very interesting as the folding mutations of DM-MBP are in the NTD and do not alter the conformational dynamics of the CTD with respect

**Fig. 4 | Three-color smFRET demonstrating the co-existence of an intermediate population and correlative refolding. a** Structure of MBP (PDB ID:1OMP; NTD in yellow, CTD in blue) showing the accessible volumes available for Atto488 (blue), Atto565 (green) and Alexa647 (red) at the labeling positions A52, K175 and P298, respectively. **b-c** Waterfall plots of FRET efficiency versus GuHCl concentration to visualize the conformational changes during equilibrium unfolding **b** and refolding **c** of triple-labeled DM-MBP. The left panels show the smFRET histograms for the NTD (BG), the middle panels show the smFRET histograms for the CTD (GR) and the right panels show the smFRET histograms for the N-C interface (BR). The white line separates the 0.9 M GuHCl measurement from the higher denaturant concentrations. **d** A comparison of the three-color smFRET histograms for molecules

measured in 0.1–0.5 M GuHCl concentrations in the native-like conformation ($E > 0.9$) for one FRET pair compared to the smFRET histograms of refolded protein for the respective FRET pair in the three-color measurement (green). The smFRET histogram of all molecules measured between 0.1 and 0.5 M GuHCl is shown in black with molecules selected with E > 0.9 are highlighted in grey. The corresponding smFRET efficiency histograms of the selected molecules for the other two FRET pairs are shown in orange. Histograms for the NTD (GR) are on the left, for the CTD in the middle and for the N-C interface on the right. Molecules selected for folded NTD are shown in the top row, for folded CTD in the middle row and for a native-like N-C interface in the bottom row.

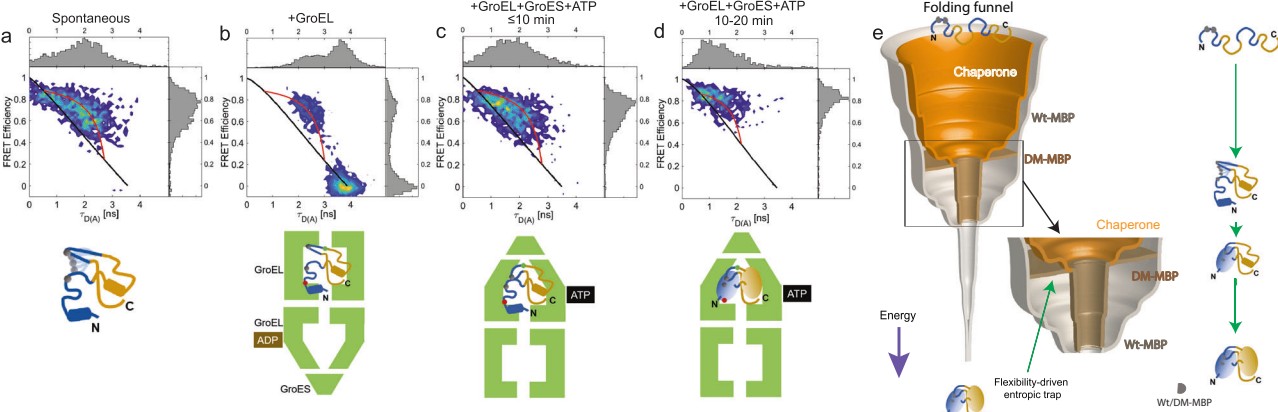

**Fig. 5 | The influence of GroEL and GroEL/ES/ATP on the folding landscape of DM-MBP. a–d** 2D FRET efficiency vs donor lifetime ($\tau_{D(a)}$) histograms (E-τ plot) of the NTD during refolding in 0.1 M GuHCl. The refolding of the NTD is shown for **a** spontaneous refolding, **b** in presence of 3 μM GroEL, and (**c–d**) upon the addition of 3 μM GroEL, 6 μM GroES and 2 mM ATP. Only the initial 10 minutes of all the three measurements are shown for (**a–c**) and between 10 and 20 min for (**d**). Above and to the right, 1-D projections are shown. The static-FRET line (black) and a dynamic-FRET line (red) are plotted for comparison. The end points for the dynamic FRET line were determined by fitting the subensemble donor fluorescence lifetimes to a biexponential. The schematic below each panel illustrates the experiment and the conformation of GroEL during the folding cycle. **e** A schematic of the protein-

folding funnel describing the refolding of WT-MBP, DM-MBP and the role of chaperones on the folding energy landscape. In the case of WT-MBP (grey), the NTD is guided by hydrophobic interactions and folds first followed by CTD folding. In the case of DM-MBP (brown), NTD folding is delayed by the high configurational entropy generated by the loss of binding energy due to less hydrophobic mutations. As soon as the NTD finds its folding competent conformation in DM-MBP, the CTD folds along the folding funnel as for WT-MBP. Chaperones shape the protein folding funnel (beige) by restricting the conformational space available to the substrate, thereby guiding the protein towards the correct conformation and thereby accelerating the refolding rate.

to the WT-MBP (Fig. S10b,d, Table S6). Still, the CTD can only finish folding after folding of the NTD, thus maintaining the folding order observed in WT-MBP.

To further investigate the origin of the differences in WT- and DM-MBP refolding, we performed all-atom molecular dynamics (MD) simulations of the temperature-induced unfolding of WT-MBP and DM-MBP starting from the native conformation (PDB: 1OMP, Supplementary Note 7 and Fig. S17). For WT-MBP, the CTD unfolds first while, for the NTD, parts of the secondary structure are preserved until the end of the simulation (Fig. S17a). Notably, these stable regions involved the residues V8 and Y283 that are mutated in DM-MBP. On the contrary, for DM-MBP, one major unfolding event was observed with no major secondary structure being preserved at the end of the simulation (Fig. S17b). This suggests that the mutations present in DM-MBP disrupt the formation of a stable folding core structure in the early stages of folding that guides the protein along the correct folding trajectory.

**GroEL/ES modulates the energy landscape of DM-MBP refolding**
As DM-MBP is a model substrate for investigating the function of bacterial chaperonin system GroEL/ES, we investigated its influence on the conformation and dynamics of DM-MBP. During the initial ten minutes of spontaneous refolding in the absence of GroEL, a broad, dynamic FRET efficiency distribution was observed for the NTD

(Fig. 5a, Fig. S18a), similarly to what we observe in the equilibrium unfolding/refolding measurements (Figs. 1c and 2b). Upon the binding of DM-MBP to GroEL, two populations are observable, an intermediate population which falls on a red dynamic FRET line and an extended or a stretched conformation on the static FRET line with a donor fluorescence lifetime of ~3.8 ns (Fig. 5b) as we have also observed previously[24]. Interestingly, the intermediate population is still dynamic, as was observed in the absence of GroEL. Strikingly, during the first ten minutes of refolding in the presence of the full GroEL/ES chaperonin system, DM-MBP is still dynamic with fluctuations between conformations similar to that observed for the free protein, although the unfolded state is more compact. This would be expected inside the cavity of GroEL/ES. However, the equilibrium is significantly shifted to the compact, native-like conformation with high FRET efficiency (Fig. 5c) from which the protein searches for the native interactions within the core of the NTD. This indicates that the compact state also exists within the chaperonin system and the shifted equilibrium enhances the time available for the search. During these measurements, we verified that DM-MBP was bound to GroEL or GroEL/ES using FCS (Fig. S18b). The stretched conformation does not exchange with the dynamic population on the millisecond timescale but disappears upon the addition of GroES and ATP. This indicates that GroEL grabs some of the substrates and extends them to remove potential deleterious non-native interactions. Upon encapsulation within the

cavity of GroEL/ES, the conformational space that the protein is allowed to explore is restricted, helping the protein to overcome the entropic barrier for refolding (Fig. 5c,d, Fig. S18a). This is consistent to what we observed before[25] and explains why GroEL/ES is able to accelerate the folding of DM-MBP by ~10-fold[24,25].

Our results are summarized in the protein folding funnel for WT-MBP, DM-MBP and GroEL/ES assisted folding of DM-MBP shown in Fig. 5e. WT-MBP has the hydrophobic core guiding folding of the NTD followed later by CTD folding[22]. The double mutations (V8G and Y283D) disrupt the nucleation core in the NTD of DM-MBP. Hence, DM-MBP has to perform a large, conformational search to find folding competent conformations. This leads to a flattening of the funnel for DM-MBP in the region where the hydrophobic interactions in WT-MBP further guide folding of the NTD (Fig. 5e). Once the correct contacts are formed in the NTD, the N-C interface and finally CTD fold. In the presence of GroEL/ES, the conformational space available is restricted, leading to an increase in the refolding rate for DM-MBP.

## Discussion

In this report, we have investigated the coordinated domain-wise refolding of DM-MBP using two-color and three-color smFRET. Our study shows that the previously known unfolding/refolding hysteresis of DM-MBP occurs under conditions where the protein undergoes rapid dynamics between extended and compact conformations on the micro- to millisecond timescale. The hysteresis cannot be attributed to aggregation effects[46] (Fig. S1) (Figs. 2–3) and suggests that dynamics may be an important phenomena to study in protein aggregation-related diseases. The MBP NTD core encompasses residues that are far apart in sequence but juxta-positioned in the native state (Fig. 1a) and form the overall rate-limiting step in protein folding. Upon disruption of this core, as is the case in DM-MBP, NTD faces an entropic barrier and is highly dynamic on the ms timescale. Refolding experiments over days in a concentration where the hysteresis is observable (0.3 M GuHCl) shows that DM-MBP can slowly refold, but also has to compete with permanent denaturation pathways (Fig. S8). As demonstrated previously, constraining the flexible region, either by introducing disulfide bridges to tether the distant segments[25] or by using TMAO (ref. 31, Fig. 2d), decreases the entropic free-energy barrier for forming native-like contacts and thereby removes the hysteresis. Although, we cannot rule out the existence of a trapped, compact state during refolding, we assume the reversibility of the fluctuations between the unfolding and compact state and use a simple folding model with an entropic barrier to estimate the free energy. The reconfiguration times during refolding were approximated from the relaxation times obtained using dynamic PDA. The resulting barrier can also be overcome by confinement in the chaperonin cage of GroEL/ES[24].

The three-color smFRET experiments reported here provide direct evidence for the folding order of the domains of DM-MBP: the NTD folds first followed by formation of the N-C interface and, as last, folding of the CTD (Fig. 4d). This folding order is consistent with the dynamic PDA analysis, which showed slow apparent rates for NTD required for contact formation and fast rates for the CTD suggesting that the CTD is still dynamic, waiting for the NTD to fold (Fig. 3b, Fig. S10, Table S6). The same folding order has been observed for WT-MBP in a recent hydrogen exchange-mass spectroscopy (HX-MS) study where the NTD folded over an order of magnitude faster ($t_{1/2}$ ~ 1 s) compared to the CTD ($t_{1/2}$ ~ 40 s)[23]. For a single-mutant variant of DM-MBP, V8G, it has been reported that the mutant significantly hinders the folding of the NTD ($t_{1/2}$ ~ 20 s) without affecting folding of the CTD ($t_{1/2}$ ~ 40 s)[22]. This implies that the dependency of the CTD on the NTD folding is conserved for the V8G single mutant. Folding of domains in another multi-domain protein has also showed a preference on the order of domain folding[11]. Until now, the influence of dynamics during the refolding process in multi-domain proteins was limited to MD simulation studies[14,15,47].

In the presence of GroEL, the hydrophobic lining of the cavity of GroEL captures the unfolded substrate. Some of the protein is stretched beyond distances observed for the unfolded protein, which helps to remove misfolded structures. This mechanism has recently been observed for other chaperone systems[48]. Other substrates still exhibit dynamic fluctuations between compact and unfolded structures. Upon the binding of GroES and ATP to GroEL, the substrate is encapsulated in the cavity and the lining of the cavity becomes hydrophilic[49]. The substrate dynamically searches for the correct structure within the limited conformational space it can explore, thereby accelerating folding (Fig. 5). These results further support the view that confinement is an effective pathway for cooperative folding and the dynamics underlying the substrate processing[4,14,47,50,51] that also prevents aggregation and minimizes misfolding stress[52,53].

As the refolding of MBP has been previously investigated, it serves as a good platform for testing the power of three-color FRET for investigating the folding order of multi-domain proteins as well as coordinated motions in general. For a complete understanding of the folding pathway of multi-domain proteins, it is also important to consider co-translational folding and the effect of chaperones that help nascent proteins attain their native structures within physiologically relevant time-scales[16,17]. This is particularly important when domains can misfold during folding[52–55] as we saw here for DM-MBP during refolding over 72 hours. In our study, we also demonstrated the importance of conformational dynamics on the ms timescale during the refolding process. Recently, mutations were used during protein designing and engineering to manipulate the folding energy landscape and gain control over allostery effects by adjusting the dynamics of the system[56]. Thus, the information gained from our two- and three-color FRET experiments can provide relevant information to researchers performing protein engineering and design. It can also be applied to obtain mechanistic insights into allosteric effects and to investigate protein-chaperone interactions, which is related to diseases involving protein aggregation.

## Methods

### MBP constructs, protein expression and purification

The plasmid encoding DM-MBP (V8G, Y283D) was a gift from F. Ulrich Hartl and Manajit Hayer-Hartl (MPI of Biochemistry, Martinsried, Germany). The DM-MBP plasmid backbone is pCH-series based, enabling IPTG inducible expression. Single cysteine (A52C, P298C) and double cysteine (A52C-P298C, K175C-P298C, A52C-K175C) DM-MBP mutants were generated using site-directed mutagenesis (Thermo Scientific-Phusion Site-Directed Mutagenesis Kit). The A52TAG-K175C-P298C mutant was generated by mutating the codon for alanine at position 52 to the amber stop codon (TAG) to incorporate the unnatural amino acid (UAA) N-Propargyl-L-Lysine (PrK).

All single and double cysteine DM-MBP mutant proteins were expressed in *E. coli* BL21-AI (L-(+)-arabinose controlled T7 RNA polymerase expressing strain, Thermo Scientific Cat. Number C607003) at 30 °C for 4 hr with the addition of both 0.2% L-(+)-arabinose and 0.5 mM IPTG in the media and under constant shaking of 200 RPM.

UAA incorporation requires the co-expression of the used orthogonal translation system of tRNA^Pyl and PylRS^WT with the expression of the given protein. For this purpose, we used pEvol-tRNA^PylPylRS^WT [57]. *E. coli* BL21-AI cells harboring both pCH-A52TAG-K175C-P298C-DM-MBP and pEvol-tRNA^PylPylRS^WT plasmids were grown for 1 hr at 30 °C before the addition of 1 mM of PrK (SiChem GmbH) in the media. PrK was prepared in a 0.1 M NaOH solution. Later, DM-MBP-A52PrK-K175C-P298C expression was achieved similar to the other mutants with L-(+)-arabinose and IPTG.

All DM-MBP proteins were purified with an amylose column (New England Biolabs) as described previously[24]. Proteins were quantified spectrophotometrically at 280 nm using an $\varepsilon_{280} = 59,000$ ODs $M^{-1}cm^{-1}$.

## Tryptophan fluorescence measurements

All tryptophan fluorescence measurements were performed at room temperature (RT, ~22 °C) on a FLS1000 Photoluminescence Spectrometer (Edinburgh Instruments). To investigate the kinetics of MBP refolding, the intrinsic tryptophan fluorescence intensity was measured for 2 s at intervals of 60 s. 3 μM MBP or DM-MBP was denatured with 3 M GuHCl in buffer A (20 mM Tris, pH 7.5, 20 mM KCl) and allowed to refold after 75-fold dilution in buffer A (yielding a final protein concentration of 40 nM). Intrinsic tryptophan fluorescence was excited at 290 nm with a slit width of 2 nm and detected at 345 nm with a 5 nm slit width. Photobleaching was minimized by adjusting the slit widths, measurement time intervals and acquisition times.

For steady-state unfolding and refolding experiments, tryptophan fluorescence was measured after 20 hr. For the unfolding curve, ~40 nM native MBP was incubated at RT in buffer A containing 0.2 M, 0.4 M, 0.6 M, 0.8 M, 1 M, 1.2 M, 1.4 M, 1.6 M, 1.8 M and 2 M GuHCl. For the refolding curve, 2 μM MBP or DM-MBP was denatured in 3 M GuHCl/10 mM DTT at 50 °C for 1 h in buffer A and incubated at RT after 50-fold dilution in 0 M, 0.2 M, 0.4 M, 0.6 M, 0.8 M, 1 M, 1.2 M, 1.4 M, 1.6 M, 1.8 M and 2 M GuHCl prepared in buffer A.

## The Boltzmann function for unfolding and refolding titrations

The steady-state unfolding and refolding titrations from the tryptophan measurements were fit using a Boltzmann function:

$$y = \frac{I_1 - I_2}{1 + e^{(c-c_0)/\Delta c}} + I_2 \qquad (2)$$

where $I_1$ is the final and $I_2$ is the initial data point respectively, $c_0$ is the center of the transition and $\Delta c$ is the increment between data points.

For equilibrium unfolding and refolding experiments with smFRET, an intermediate population was observed. In these cases, the titrations were fit using a double Boltzmann function given by:

$$y = I_2 + \frac{(I_1 - I_2)f_1}{1 + e^{(c-c_{0,1})/\Delta c}} + \frac{(I_1 - I_2)(1-f_1)}{1 + e^{(c-c_{0,2})/\Delta c}} \qquad (3)$$

where $f_1$ is the fractional amplitude of the first transition, $c_{0,1}$ and $c_{0,2}$ are the centers of the first and second transitions and $\Delta c$ is the respective increment between data points.

## Fluorophore labeling of WT- and DM-MBP

**Cysteine-maleimide labeling with one and two fluorophores.** All cysteine-maleimide reactions were performed according to the manufacturer's instructions (Atto-Tec) with a few modifications. The single-cysteine mutant, A52 C was labeled either with Atto532- (Atto-Tec) or Alexa647-maleimide (Invitrogen). All double-cysteine mutants, A52C-P298C, K175C-P298C and A52C-K175C were stochastically labeled with Atto532- and Alexa647-maleimides. To analyze two color controls measurements for the three-color FRET analysis, the double-cysteine mutant A52C-P298C was stochastically labeled with Atto488- and Atto565-maleimides, K175C-P298C with Atto565- and Alexa647-maleimides, and A52C-K175C with Atto488- and Alexa647-maleimides.

Briefly, the sulfhydryl groups of ~50 μM WT- or DM-MBP cysteine mutants were reduced with 10 mM DTT in phosphate buffered saline solution (PBS) at RT for 20 min. Excess DTT was removed by washing the protein in a 10 kDa cut-off Amicon centrifugal filter (Merck-Millipore) with de-oxygenated PBS containing 50 μM tris(2-carboxyethyl) phosphine (TCEP). Approximately a 3-fold molar excess of the maleimide-fluorophore conjugate was added to the washed protein solution and the reaction was carried out at RT for 3 hr in the dark. For double labeling, the fluorophores were labeled stochastically by adding an equimolar mixture of both maleimide-fluorophore conjugates simultaneously in the reaction. Unreacted maleimide-fluorophores were washed out with buffer A containing 1 mM DTT using a

centrifugal filter. Successful labeling was verified using FCS. FCCS was used to confirm double-labeling. The degree of double labeling was found to be between 15-25% when quantified spectrophotometrically. Labeling of WT- or DM-MBP with the above-mentioned cysteine-maleimide chemistries did not affect the refolding rates (Table S1).

**Specific labeling with three fluorophores.** DM-MBP-A52PrK-K175C-P298C protein was specifically labeled with three fluorophores for the three-color smFRET experiments. We used the dyes Atto488, Atto565 and Alexa647, which were chosen to maximize the use of visible spectrum with a distance sensitivity indicated by their respective Förster distances (~50-70 Å) (Table S2). In the first step, A52PrK with an alkyne group was specifically conjugated to the azide moiety of Atto488-azide (Atto-tec) via copper-catalyzed alkyne-azide cycloaddition, one type of click chemistry reaction[58]. Approximately 120 μM of DM-MBP-A52PrK-K175C-P298C protein was allowed to react with 3-fold molar excess of Atto488-azide in the presence of 200 μM CuSO$_4$, 50 μM TCEP, 200 μM Tris(benzyltriazolylmethyl)amine and freshly prepared 200 μM sodium ascorbate in PBS at RT for ~3 hr, in the dark, under mild shaking conditions. Unreacted dye was removed by washing with PBS using centrifugal filters (Amicon, 10 kDa cutoff).

K175C and P298C were labeled with Alexa647 and Atto565, respectively, using cysteine-maleimide chemistry. It has been shown that maltose binding in the inter-domain cleft of MBP buries some residues at the domain-interface including position 298[24]. Therefore, we labeled the Alexa647-maleimide specifically to the cysteine at 175 position by labeling in the presence of maltose, which blocks the competing cysteine at position 298[24,59,60]. For the second step of labeling, we took ~70 μM of 52PrK-atto488 labeled DM-MBP-A52PrK-K175C-P298C, reduced it with 10 mM DTT addition and washed with de-oxygenated PBS containing 500 mM Maltose and reconcentrated using the centrifugal filters. The cysteine-maleimide reaction was performed by the addition of 2-fold molar excess of Alexa647-maleimide and allowed to react for 1 hr at RT. Under these conditions, the overall labeling efficiency of position K175C is 40%. However, more important than the labeling efficiency at this point is minimizing the possibility of mis-labeling the maltose blocked cysteine at position 298 with Alexa 647 (Fig. S12). Molecules that lack all three dyes are not included in further three-color analyses, whereas molecules with the dyes labeled unspecifically would complicate the analysis. As a third and final step, ~30 μM of 52PrK-Atto488-175C-Alexa647 labeled DM-MBP-A52PrK-K175C-P298C protein was washed with PBS and reconcentrated with centrifugal filters to remove the excess of unlabeled Alexa647 dye and maltose. The washed protein was then labeled with 3-fold molar excess of Atto565-maleimide to the only available cysteine at position 298. Coupling of each dye after each labeling step was monitored by measuring the absorption of the respective fluorophores at their respective wavelengths of maximum absorption and of the protein at 280 nm. The overall degree of labeling for all three labels was estimated to be ~15% by absorption spectroscopy. However, in the burst analysis experiment, a lower limit ~2% of the bursts were attributed to triple-labeled molecules with sufficient statistics to analyze the FRET histograms after filtering for photobleaching and blinking events (Fig. S13b-c). Covalent attachment of the three-fluorophores did not have any significant influence on protein functionality (Fig. S14) and refolding (Fig. S13f, Fig. S15d, e).

**Single molecule FRET measurements.** All the single-molecule FRET measurements were performed on custom-built confocal set-ups as described below. 50-100 pM of double- or triple-labeled DM-MBP proteins were measured to minimize the possibility of having more than one molecule in the confocal volume at a time. Before starting a smFRET measurement, the surface of the sample holder (LabTek I 8 chamber slides, VWR) was passivated with 1 mg/ml BSA. SmFRET measurements of native MBP proteins were performed after serially

diluting the labeled proteins in buffer A (20 mM Tris, pH 7.5, 20 mM KCl) to the desired concentration range. To measure the FRET efficiency in the completely unfolded state, first ~500 nM labeled protein was denatured in buffer A containing 3 M GuHCl and 10 mM DTT at 50 °C for 1 h and later was measured after serially diluting the protein concentration to 50-100 pM in 2 or 3 M GuHCl prepared in buffer A. SmFRET experiments performed in 6 M GuHCl showed no additional changes in the conformation of the denatured protein (data not shown).

For all unfolding and refolding smFRET measurements, 0.001% tween-20 was added to the buffer to prevent unfolded and refolded molecules from sticking to the surface. To measure the FRET efficiency under unfolding conditions, 10-20 μM labeled native protein was diluted to 50–100 pM in buffer A containing either 0 M, 0.1 M, 0.2 M, 0.3 M, 0.5 M, 0.9 M, 1 M, 2 M or 3 M GuHCl. Refolding smFRET measurements were performed by first denaturing 500-1000 nM protein in 3 M GuHCl and 10 mM DTT at 50 °C for 1 hr followed by serial dilutions in 3 M GuHCl and a final 50-fold dilution in buffer A with the appropriate GuHCl concentration to obtain 50-100 pM labeled protein in solution. For refolding measurements of the NTD construct of DM-MBP over 3 days in 0.3 M GuHCl, the reaction was initiated by diluting the 3 M GuHCl denatured DM-MBP to a protein concentration of 40 nM. Small volumes were further diluted to 50-100 pM concentrations immediately before the individual smFRET measurements while keeping the final GuHCl concentration at 0.3 M. The FRET measurements were performed from the same reaction mixture after 0, 18, 48 and 70 hr from the start of the reaction. For temperature series measurements, 25 °C, 30 °C, 35 °C temperatures were controlled using a heated stage and objective heater equilibrated for 10 minutes before the start of the measurement.

For the measurements with GroEL and GroEL/ES/ATP, refolding of the NTD was performed in 0.1 M GuHCl in the presence of 3 μM GroEL, and upon the addition of 3 μM GroEL, 6 μM GroES and 2 mM ATP. Each measurement was for 10 minutes. The experiment was repeated 5-10 times and combined together to increase the statistics.

**Two-color setup.** Fluorescence correlation spectroscopy, fluorescence cross-correlation spectroscopy and two-color smFRET measurements on WT- and DM-MBP proteins labeled with Atto532 and Alexa647 were performed on a home-built confocal microscope capable of multi-parameter fluorescence detection (MFD) combined with pulsed interleaved excitation (PIE, MFD-PIE). PIE was implemented using a 532 nm green laser (Toptica; PicoTA 530) to excite the donor and a 640 nm red laser (PicoQuant; LDH-D-C-640) for directly exciting the acceptor molecules. The lasers were synchronized at a repetition rate of ~26.7 MHz with a delay of ~18 ns between each pulse. Both lasers were combined into a single-mode optical fiber using a custom-designed laser combiner (AMS Technologies, WDM-12P-11-532-/640-3.5/125-PPP-50-3A3A3A-3-1,1,2) to clean up the beam profile and the beam diameter exciting the fiber was adjusted to ~2 mm using a collimator (0FC-4-RGB11-47, Schäfter + Kirchhoff, Germany), underfilling the objective to increase the size of the excitation volume. Laser excitation powers were set to ~100 μW before the objective for both the lasers. A 60x water immersion objective (Nikon; Plan Apo IR 60×1.27 Water Immersion) was used to collect the emitted fluorescence and focused onto a 75μm diameter pinhole for confocal detection. To implement MFD, the fluorescence signal collected after the pinhole was separated into parallel and perpendicular polarized light by a polarizing beam-splitter (Thorlabs; PBS3) and then spectrally separated for green and red fluorescence detection by a dichroic mirror (AHF Analysetechnik; Dual Line z532/635,). Finally, emission filters (green: Semrock, Bright line 582/75; red: Chroma, HQ700/75 M) were placed before the four single-photon-counting (SPC) avalanche photodiodes (APD) (Perkin-Elmer) used for detection. Four independent but synchronized SPC cards (Becker and Hickl; SPC 154) were synchronized with the laser drivers to record the arrival times of the photons.

**Fluorescence Correlation and Cross-correlation Spectroscopy.** Fluorescence correlation spectroscopy (FCS) and fluorescence cross-correlation spectroscopy (FCCS) experiments were performed with PIE capabilities to investigate aggregation of DM-MBP during refolding[61]. For these experiments, a single-cysteine mutant of DM-MBP (A52 C) labeled with either Atto532 maleimide or Alexa647 maleimide dye was used and measured with the two-color MFD-PIE setup as described above. FCCS detects the coincidence of fluctuations in both the green and red detection channels and hence can sensitively detect the presence of oligomers. For these experiments, 500 nM Atto532-DM-MBP and 500 nM Alexa647-DM-MBP were denatured in 3 M GuHCl at 50 °C for 1 hr. The samples were then mixed at RT and refolding was initiated by diluting the solution to a final labeled-protein concentration of 40 nM (20 nM of Atto532-DM-MBP and 20 nM of Alexa647-DM-MBP) in buffer A. FCCS experiments were performed at 0, 30 and 60 minutes after dilution to monitor the time-dependent oligomer formation over the entire refolding process. A 40 base-pair dsDNA labeled with both Atto532 and Atto647 on different strands was used as a positive control. Free dyes were measured as a negative control. Native and denatured proteins were also assayed for aggregation. FCCS analysis of DM-MBP refolding measurements performed in 0.3 M GuHCl for three days was also analyzed as mentioned above after 0, 18, 48 and 70 hr at a total concentration of 40 nM (Fig. S8d).

The confocal volume was approximated as a 3D Gaussian yielding the following analytical equation for the auto-correlation function:

$$G(\tau) = \frac{\gamma}{N} \left(1 + \frac{\tau}{\tau_D}\right)^{-1} \left(1 + \frac{\tau}{\tau_D}\frac{1}{\rho^2}\right)^{-\frac{1}{2}} + y_0 \qquad (4)$$

where, $\gamma = 2^{-3/2}$ is the geometric factor used to correct for confocal shape, $N$ is the average number of diffusing molecules in the probe volume with a diffusion time $\tau_D$:

$$\tau_D = \frac{w_0^2}{4D} \qquad (5)$$

where $D$ is the diffusion coefficient, $\rho$ is the structure parameter defined as $w_0/z_0$, where $w_0$ and $z_0$ are the axial and radial dimensions from the center of the point-spread-function to the position where the intensity has decayed to $\frac{1}{e^2}$ and $y_0$ is a baseline to compensate for a potential offset in the correlation functions.

For the FCCS analysis, the amplitudes of the green and red auto-correlation functions can be used to determine the total number of diffusing particles containing both green and red labels:

$$N_{GT} = N_G + N_{GR} \qquad (6)$$

$$N_{RT} = N_R + N_{GR} \qquad (7)$$

where $N_G$ and $N_R$ represent the number of diffusing particles containing only a green and a red label respectively. The number of double-labeled molecules ($N_{GR}$) was determined from the amplitude of the cross-correlation function, which is given by:

$$G_{CC}(0) = \gamma \frac{N_{GR}}{N_{GT} N_{RT}} \qquad (8)$$

**Two-color MFD-PIE analysis.** For two-color smFRET measurements, ~50-100 pM double-labeled MBP sample was measured. A MFD-PIE data analysis was performed to calculate the correct two-color FRET

efficiency, burst-wise lifetimes and anisotropy for single molecule events (Fig. S2) as described previously[27,62,63]. Single-molecule bursts in two-color FRET measurements were distinguished from the background using a photon burst search algorithm by applying a threshold of at least 5 photons for sliding time window of 500 μs and a total of 50 photons per burst[64]. The burst-wise fluorescence lifetime was estimated from the fluorescence decay by convolution with the instrument response function. The fluorescence anisotropy as a function of fluorescence lifetime was fitted using the Perrin equation as discussed previously[65]:

$$r = \frac{r_0}{1 + \frac{\tau}{\rho}} \tag{9}$$

where, $r$ is steady state anisotropy, $r_0$ is the fundamental anisotropy, $\rho$ is rotational correlation time and $\tau$ is a fluorescence lifetime. Molecules labeled with both donor and acceptor dyes, which show a stoichiometry of ~0.5, were selected for further analysis. An ALEX-2CDE filter with an upper value of 12 was used to filter out photobleaching and blinking events[66]. After correcting for background, crosstalk of green fluorescence in red channel ($\alpha$), direct excitation of acceptor by donor excitation laser ($\delta$), and differences in detection efficiencies and quantum yields of the dyes ($\gamma$) were estimated. The corrected labeling stoichiometry ($S$) and FRET efficiency ($E$) for all the bursts was calculated as:

$$S = \frac{\gamma F_{GG} + F_{GR} - \alpha F_{GG} - \delta F_{RR}}{\gamma F_{GG} + F_{GR} - \alpha F_{GG} - \delta F_{RR} + F_{RR}} \tag{10}$$

$$E = \frac{F_{GR} - \alpha F_{GG} - \delta F_{RR}}{\gamma F_{GG} + F_{GR} - \alpha F_{GG} - \delta F_{RR}} \tag{11}$$

where, $F_{GG}$ and $F_{GR}$ are the background-corrected fluorescence intensities in the donor and acceptor channel after donor excitation respectively and $F_{RR}$ is the background-corrected fluorescence signal in red channel after acceptor excitation. For the determined correction factors, see Table S2.

Ideally, when the conformation of the protein is static while the molecule transits the laser spot, the FRET efficiency (e) is related to the fluorescence lifetime of the donor in the presence of an acceptor as:

$$E = 1 - \frac{\tau_{D(A)}}{\tau_{D(0)}} \tag{12}$$

where $\tau_{D(A)}$ is the fluorescence lifetime of the donor in the presence of an acceptor and $\tau_{D(0)}$ is the fluorescence lifetime of the donor in the absence of an acceptor. This relationship changes slightly when the inter-dye separation becomes comparable to the relative linker lengths, where linker flexibility dominates. An accurate relationship can be derived for a specific set of fluorophores as has been described previously[63]. For the dye pair Atto532-Alexa647, this relationship is given by the following third-order polynomial:

$$E = 1 - \frac{-0.0178 + 0.6226\langle\tau_{D(A)}\rangle + 0.2188\langle\tau_{D(A)}\rangle^2 + 0.0312\langle\tau_{D(A)}\rangle^3}{\langle\tau_{D(0)}\rangle} \tag{13}$$

When dynamics are present between two states with their respective donor fluorescence lifetimes $\tau_1$ and $\tau_2$, the relationship of intensity averaged FRET efficiency ($E$) to the donor lifetime changes to[64]:

$$E = 1 - \frac{\tau_1 \cdot \tau_2}{\tau_{D(0)}[\tau_1 + \tau_2 - \langle\tau\rangle]} \tag{14}$$

where the average donor fluorescence lifetime $\langle\tau\rangle$ is calculated from the total florescence signal over a burst.

## Fluorescence lifetime analysis

To determine the FRET efficiencies of the different FRET states undergoing dynamic transitions, a fluorescence lifetime analysis was performed. The photons from all selected, dual-color bursts were sum together and fit to a biexponential function convoluted with the instrument response function and a scattering component was included as an additional species in the fit. From the fluorescence lifetime, the FRET efficiency and distances were calculated using:

$$E = 1 - \frac{\tau_{D(A)}}{\tau_{D(0)}} \tag{15}$$

and

$$E = \frac{1}{1 + \left(\frac{R}{R_0}\right)^6} \tag{16}$$

where $\tau_{D(A)}$ is the fluorescence lifetime of donor in the presence of an acceptor, $\tau_{D(0)}$ is the fluorescence lifetime of the donor in the absence of an acceptor (~3.6 ns for Atto532), $R$ is the distance between the dyes and $R_0$ is the Förster distance (62 Å for the Atto532-Alexa647 dye pair). The quality of the fit was evaluated using the reduced $\chi^2$ value, $\chi^2_{red}$.

## Three-color setup

Both triply-labeled DM-MBP with Atto488-Atto565-Alexa647 and doubly-labeled DM-MBP with Atto488-Atto565, Atto565-Alexa647 and Atto488-Alexa647 dye pairs were measured on a three-color confocal single-molecule setup equipped with MFD-PIE as described previously[67]. Briefly, PIE experiments were performed with three pulsed lasers having ~20 ns delay between each pulse (PicoQuant, Germany; LDH-D-C-485, LDH-D-TA-560, LDH-D-C-640). The pulse frequency of 16.7 MHz and its synchronization were achieved using a laser driver (PicoQuant, Germany; Sepia II). A 60× water immersion objective with 1.27 N.A. (Nikon, Germany; Plan Apo IR 60×1.27 WI) was used to focus the lasers into the sample with a power measured before the objective of ~120 μW for blue, ~75 μW for green and ~35 μW for the red laser. The emitted fluorescence was collected by the same objective and separated from the excitation beam using a polychroic mirror (AHF Analysentechnik; zt405/488/561/633, Germany) and passed through a 50 μm pinhole for defining the confocal volume. Light coming through the pinhole was first separated into its parallel and perpendicular polarization components with a polarizing beam splitter (Thorlabs, Germany; PBS251). Afterwards, the light in each polarization channel was separated into blue, green and red spectral regions using two dichroic mirrors (AHF Analysentechnik; BS560, 640DCXR). The blue, green and red detection channels were spectrally defined using emission filters (AHF Analysentechnik; ET525/50, ET607/36, ET670/30) before the fluorescence was detected on APD's (Laser-Components, 2x COUNT-100B; Perkin Elmer, 4x SPCM-AQR14). The timing of the detected photons was synchronized with the lasers pulses using a TCSPC module (PicoQuant; HydraHarp400).

## Three-color MFD-PIE analysis

In a three-color FRET measurement, the blue dye acts as a donor (d), the green dye acts as a first acceptor ($A_1$) and the red dye as a second acceptor ($A_2$). Additionally, the green dye can also act as a donor for the red dye (Figure S13a). Extending MFD-PIE to three colors makes it possible to detect photons in the green and red channels after blue excitation as well as in the green and red channels after green

excitation. This enables one to calculate all three stoichiometries ($S_{BG}$, $S_{BR}$ and $S_{GR}$) and FRET efficiencies ($E_{BG}$, $E_{BR}$ and $E_{GR}$) for blue-green, blue-red and green-red dye-pairs respectively. The latter case for the green-red dye pair is similar to the typical two-color MFD-PIE scheme.

For three-color FRET measurements, ~50-100 pM triple labeled DM-MBP was measured on a passivated glass surface. An all-photon burst search algorithm was used to detect the single-molecule events from the background and required at least 30 photons per sliding window of 500 μs and a total of 100 photons per burst. A typical burst-wise MFD-PIE analysis including stoichiometry, FRET efficiency, fluorescence lifetime and anisotropy, was extended to three-color MFD-PIE on the selected bursts as described previously (Fig. S13d-e)[25].

Briefly, the three stoichiometries ($S_{BG}$, $S_{BR}$ and $S_{GR}$) were calculated as follows:

$$S_{BG} = \frac{F_{BB} + F_{BG}}{F_{BB} + F_{BG} + F_{GG}} \tag{17}$$

$$S_{BR} = \frac{F_{BB} + F_{BR}}{F_{BB} + + F_{BR} + F_{RR}} \tag{18}$$

$$S_{GR} = \frac{F_{GG} + F_{GR}}{F_{GG} + F_{GR} + F_{RR}} \tag{19}$$

where $F_{XY}$ represents the detected fluorescence signal in the $Y$ channel after exciting with the $X$ laser. For the three-color analysis, triple labeled molecules were sorted by applying the ALEX-2CDE-filter for all the three stoichiometries with a maximum value of 15 for both blue-green, blue-red and 20 for green-red dye-pairs. Typical values of $S_{BG}$, $S_{BR}$ and $S_{GR}$ for triple-labeled DM-MBP molecules with Ato488, Atto565 and Alexa647 dyes were found to be ~0.2, ~0.15 and ~0.5 respectively (Fig. S13b-c).

The corrected three FRET efficiencies for the selected triple-labeled molecules were derived as detailed in Barth et al. 2019[67], and can be written as:

$$E_{GR} = \frac{F_{GR} - \alpha_{GR} F_{GG} - \delta_{GR} F_{RR}}{\gamma_{GR} F_{GG} + F_{GR} - \alpha_{GR} F_{GG} - \delta_{GR} F_{RR}} \tag{20}$$

$$E_{BG} = \frac{F_{BG}^{cor.}}{\gamma_{BG} F_{BG}(1 - E_{GR}) + F_{BG}^{cor.}} \tag{21}$$

$$E_{BR} = \frac{F_{BR}^{cor.} - E_{GR}\left(\gamma_{GR} F_{BG}^{cor.} + F_{BR}^{cor.}\right)}{\gamma_{BR} F_{BB} + F_{BR}^{cor.} - E_{GR}\left(\gamma_{BR} F_{BB} + \gamma_{GR} F_{BG}^{cor.} + F_{BR}^{cor.}\right)} \tag{22}$$

where the intermediate correction terms $F_{BG}^{cor.}$ and $F_{BR}^{cor.}$ are defined as:

$$F_{BG}^{cor.} = F_{BG} - \alpha_{BG} F_{BB} - \delta_{BG} F_{GG} \tag{23}$$

$$\begin{aligned} F_{BR}^{cor.} = F_{BR} - \alpha_{BR} F_{BB} - \delta_{BR} F_{RR} - \alpha_{GR}(F_{BG} - \alpha_{BG} F_{BB}) \\ - \delta_{BG} E_{GR}(1 - E_{GR})^{-1} F_{GG} \end{aligned} \tag{24}$$

The respective crosstalk, direct excitation and detection correction factors are depicted as $\alpha_{XY}$, $\delta_{XY}$ and $\gamma_{XY}$ for signal in channel $Y$ after excitation with the $X$ laser.

## Dynamic photon distribution analysis (dynamic PDA)

The raw photon signal carries important information about the kinetic heterogeneity of the system. For the purpose of computing the interconversion rates between two states in a robust way, first the proximity ratio (*PR*) collected during the burst was sliced into 0.5, 1 and

1.5 ms time bins to capture the influence of the kinetics.

$$PR = F_{GR}/(F_{GR} + F_{GG}) \tag{25}$$

A global analysis of all three-time bins was performed to extract the rates. To take care of the broadening due to photon detection noise, a constant width ($\sigma$) for a static state was assumed to scale with the inter-dye distance $R$ (i.e. $\sigma = 0.07R$)[65]. The states were defined using the donor fluorescence lifetimes of the double-labeled molecules (See Table S4). Additional states were incorporated to account for impurities and donor only molecules in the sample visible at low proximity ratios. The analysis was applied to extract the kinetic rates between the unfolded and compact states in DM-MBP during refolding in various denaturant concentrations and are reported in Table S6.

## Filtered fluorescence correlation spectroscopy analysis

A filtered fluorescence correlation spectroscopy (fFCS) analysis was performed on the smFRET refolding measurements in 0.2, 0.3, 0.5, 0.9 M GuHCl for all the three two-color constructs of DM-MBP (NTD, CTD and the N-C interface). Details of the analysis are described previously[68,69]. Briefly, two sub-populations were selected using low and high FRET efficiency thresholds for defining the individual species. For both the NTD and CTD constructs, we used $E \le 0.5$ and $E \ge 0.9$ FRET efficiency thresholds for refolding measurements in 0.2, 0.3 M GuHCl and $E \le 0.25$ and $E \ge 0.6$ for measurements in 0.5 and 0.9 M GuHCl. For the N-C interface, $E \le 0.7$ and $E \ge 0.9$ thresholds were used for refolding measurements in 0.2, 0.3, 0.5 M GuHCl, and $E \le 0.5$ and $E \ge 0.9$ thresholds were used for the measurement in 0.9 M GuHCl. The filters were generated for both donor and donor sensitized acceptor channels for the above two sub-populations after combining their micro-time patterns. A buffer component was also included and determined from the scatter and background signal of a buffer measurement. The correlation functions were calculated for the double-labeled bursts such that 50 ms before and after the end of the each burst signal is included in the analysis. After-pulsing effects of the detectors were avoided by cross-correlating the signal in both available parallel and perpendicular channels for both the donor and donor sensitized acceptor signals. The two species autocorrelation functions (SACF) and the two species cross-correlation functions (CCF) were globally fitted with a model including normal diffusion and two kinetic terms as given below:

$$G(\tau) = G_{\text{diff}}(\tau)\, G_{\text{kin}}(\tau) \tag{26}$$

$$G_{\text{diff}}(\tau) = \frac{\gamma}{N} \frac{1}{\left(1 + \frac{\tau}{\tau_D}\right)} \frac{1}{\sqrt{\left(1 + \frac{\tau}{\tau_D}\frac{1}{\rho^2}\right)}} + y_0 \tag{27}$$

$$G_{\text{kin}}^{\text{SACF}}(\tau) = 1 + A_1 e^{-\frac{\tau}{\tau_{R1}}} + A_2 e^{-\frac{\tau}{\tau_{R2}}} \tag{28}$$

$$G_{\text{kin}}^{\text{CCF}}(\tau) = 1 - A_1 e^{-\frac{\tau}{\tau_{R1}}} - A_2 e^{-\frac{\tau}{\tau_{R2}}} \tag{29}$$

Here, the $G_{\text{diff}}(\tau)$ term is the same as mentioned above in the FCS section. $G_{\text{kin}}(\tau)$ is the kinetic part of the correlation function. $\tau_{R1}$ and $\tau_{R2}$ are the two kinetic relaxation times obtained after globally fitting the four correlation functions with their respective amplitudes $A_1$ and $A_2$. For each construct, all the four correlations were globally optimized for all the refolding measurements to obtain the optimum relaxation times.

## Data analysis software

All the fluorescence correlation, burst analysis, fluorescence lifetime and PDA were performed with the open-source PIE Analysis with

**Table 1 | Parameters used in MD simulations**

| Molecular Dynamics Simulation Parameters | |
|---|---|
| Simulation Box Dimensions | 3 nm octagonal box around the molecule (PDB: 1OMP) |
| Approximate Volume | $1567 \pm 6$ nm$^3$ |
| Total number of atoms (including solvent molecules) | 152688 |
| Total number of water molecules: | 146952 atoms or 48984 H$_2$O molecules |
| Salt concentration: | 13 Na$^+$, 4 Cl$^-$ (~4 mM of excess salt) |

MATLAB (PAM) software, a custom-written, publicly available software in MATLAB (The MathWorks)[70].

## All-atom molecular dynamics simulations

All-atom molecular dynamics (MD) simulations of unfolding were performed for both WT- and DM-MBP (for details, see Table 1). For simulations on WT-MBP, the crystal structure with Protein Data Bank (PDB) ID 1OMP was used[26]. For DM-MBP, the amino acids at position 8 and 283 were exchanged with glycine and aspartate (V8G, Y283D) respectively to create the DM-MBP structure in PyMOL (Version 2.0 Schrodinger)[71]. The AMBER16 MD package with the ff14SB force field was used for the simulations[72]. A TIP3P water model was used to solvate the MBP molecule in a box of octagonal geometry. Care was taken to exclude the vaporization effects at high temperature[73]. An initial distance of 3 nm between the protein and the walls of the box was chosen to avoid boundary effects in the simulations upon unfolding. The charge of the system was neutralized by the addition of sodium ions. The protonation state of amino acids was assigned automatically using LeAP considering optimal hydrogen bonding following standard protonation states. A small excess of sodium chloride was added, corresponding to a concentration of ~4 mM. The system was allowed to equilibrate at 298 K before heating the system to 400/450 K. The stability of system was verified at high temperatures of 400/450 K during the initial 500 ps of equilibration using 2 fs steps. Production runs were performed using the NPT ensemble on a Nvidia GTX 1080 Ti GPU, running at an average of 50 ns per day. The unfolding trajectory was analyzed using AmberTools[74] by performing a secondary structure assignment using the DSSP (Definition of Secondary Structure of Proteins) algorithm in time steps of 5 ns[75]. We performed unfolding simulations at 400 K and 450 K. Simulations at 400 K were run for 2 μs to check the stability of the system at high temperature. Later, simulations were continued for another 2 μs at 450 K.

## Accessible volume calculations

To model the FRET distances in the labeled DM-MBP for comparison with the experimentally determined distances of the various labeling positions (PDB ID: 1OMP), we performed geometric accessible volume (AV) calculations using the FRET positioning and screening (FPS) software[29]. The input parameters used for simulating the dye with the AV1 model were: 20 Å (dye linker-length), 4.5 Å (dye width), and 3.5 Å (dye radius)[76–78].

## Reporting summary

Further information on research design is available in the Nature Portfolio Reporting Summary linked to this article.

## Data availability

Raw data for the main figures Figs. 1b-c, 2a, c-d, 4b-c, 5b, c-d and for the supplementary figures Fig. S1a-b, Fig. S3c, e, g, Fig. S5, Fig. S7a-b, Fig. S8a, d, Fig. S9, and Fig. S15a-c, Fig. S17 are deposited on zenodo https://doi.org/10.5281/zenodo.8007031 (part I) and https://doi.org/10.5281/zenodo.8136592 (part II). PDB ID used:1OMP. Source Data for

the main figures are provided as a Source Data file. Source data are provided with this paper.

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

## Acknowledgements

We thank Prof. F. Ulrich Hartl from the Max Planck Institute of Biochemistry, Martinsried, for kindly providing the DM-MBP plasmid and GroEL/ES proteins. We thank both Prof. F. Ulrich Hartl and Prof. Ben Schuler for valuable discussions. We gratefully acknowledge funding from the German Research Foundation (DFG) via the Sonderforschungsbereich 1035 (Projekt number 201302640, project A11 to DCL.) This work was also supported by the Federal Ministry of Education and Research (BMBF) and the Free State of Bavaria under the Excellence Strategy of the Federal Government and the Länder through the ONE MUNICH Project Munich Multiscale Biofabrication. We thankfully acknowledge the support of the Ludwigs-Maximillians-Universität München through the Center for NanoScience (CeNS) and LMUinnovativ BioImaging Network (BIN).

## Author contributions

G.A. and D.C.L. conceived the project. G.A. prepared and labeled the samples, and performed the 2 C and 3 C FRET measurements. AB performed the molecular dynamic simulations. Both G.A. and A.B. analyzed the data and prepared figures. D.C.L. acquired funding and supervised the project. G.A. wrote the original draft and all authors were involved in editing and finalizing the manuscript.

## Funding

## Competing interests

The authors declare no competing interests.
