## [Peer Review File · Nature Communications]

Reviewer #1

(Remarks to the Author)

The folding of large multi-domain proteins is understudied compared with the extensive work that has been done on small single domain proteins. For those multidomain proteins whose folding has been studied, proteins with a sequential "beads on a string" domain topology are the best characterized, while proteins with discontinuous, interdigitated, domains have received less attention.

This manuscript presents a very thorough study of the folding of a multidomain protein with a complex, discontinuous topology using multiple techniques. As such, it addresses an understudied aspect of protein folding.

I have a few questions.

In figure S3 (D, F, and H), the authors present refolding kinetics for the N and C terminal domains and the interface. It appears that all 3 regions fold on very similar timescales, and that the C terminal domain actually has a slightly shorter refolding $t_{1/2}$ than the N terminal domain. This seems inconsistent with the finding that the C terminal domain requires prior formation of the N terminal domain to fold. Do the authors have a proposed explanation for this?

On page 8 the authors write:

"It has been suggested earlier that DM-MBP refolding must overcome an entropic barrier (25) meaning that, in the absence of the hydrophobic core, a wide conformational search on a flat energy surface needs to be performed. If this were the case, we would expect refolding of DM-MBP at higher GuHCl concentrations to occur on longer timescales."

I am puzzled by the reasoning here. Denaturants are generally observed to slow the kinetics of protein refolding, regardless of the specific folding mechanism.

Reviewer #2

(Remarks to the Author)

In this work, Agam et al. study the folding of a 2-domain protein DM-MBP, in isolation as well as in the presence of the GroEl/ES chaperone. Detailed studies of multidomain protein folding lag in the field, even though such proteins represent a majority of eukaryotic proteomes. In the current work, several powerful features of multiparameter single molecule fluorescence/FRET including 3-color experiments are leveraged nicely in the study. Interesting insights emerge, including presence of hysteresis, involvement of an entropic barrier and mechanistic understanding of the accelerated folding in the chaperone by reduction of the entropic barrier due to confinement. Especially exciting is the use of filtered FCS and 3-color smFRET to obtain direct information about the order of domain folding. The combination of new insights on a multidomain protein system and interesting application of various powerful features of multiparameter single molecule fluorescence will make this work of substantial interest. However, there are some issues that need to be addressed.

The authors discuss based on their data that the observed intermediate state is due to rapid fluctuations between folded and unfolded protein states (e.g., page 8). Presumably that authors are saying that that this observed "intermediate state" is not a separate intermediate protein state, but rather a result of the measurement timescale being slower than the protein transition kinetics (the latter is discussed in the paper based on the data presented and is also a well-known kinetic phenomenon in the smFRET field including in the authors' work). Authors, please clarify the use of the "intermediate state" terminology/address as needed.

On page 18, it is unclear which data are showing the stretched conformation referred to in the text.

The authors discuss that the work shows that "the unfolding/refolding hysteresis of DM-MBP occurs

due to rapid dynamics between extended and compact conformations on the micro- to millisecond timescale". Authors please clarify how such rapid fluctuations may be giving rise to hysteresis on an orders of magnitude slower timescale.

In the labeling for the 3 color smFRET experiments, do the authors have data supporting site specific labeling? In particular, while the strategy of blocking Cys 298 with maltose is neat, if the labeling efficiency is not high, the subsequent labeling step with Atto 565 maleimide will react with the remaining unlabeled Cys 175 in addition to Cys 298. Such a mixture might contribute to/complicate the smFRET data, in particular to the BG component. Authors please address.

On page 5, a nice FCCS control is used to rule out aggregation. The authors may choose to mention the experiment briefly in the main text for readers' benefit.

For Figure 1B discussion of hysteresis on page 5, it is worth noting the time delay after making the sample for the experiment.

Presumably the N-C mutant combines FRET information about the interface and folding of NTD/CTD. Authors, please briefly comment on this point.

Reviewer #3

(Remarks to the Author)

In this manuscript, Agam and co-workers studied the folding of DM-MBP, an MBP variant with reduced folding rates, using smFRET spectroscopy. In their study they monitor folding of the C-terminal domain, the N-terminal domain and the overall protein using FRET pairs on different locations. They further perform three colour FRET experiments to simultaneously monitor DM-MBP folding and also monitor folding in presence of GroEL/ES. They find that the major folding event is limited by an entropic barrier, which is still present in GroEL/ES but reduced.

This manuscript is from the smFRET side rigorously executed. Fluorophore positions are carefully chosen and the data is of high quality and analysed to recently defined standards in the field. The biological model system DM-MBP is very well established and very well studied. This is a point, where I was struggling most, because it has not become clear to me where the real novelty of the study is. These mutations are known for a long time and the kinetic trapping as well during the folding from ensemble experiments, therefore I see here limited new insights - as also indicated by the. discussion section of the manuscript.

One of the major challenges in the model system is in my eyes the reversibility. The authors show already in Fig 1B that even though performed presumably at equilibrium, the unfolding and refolding TRP curves do not overlap but show a huge hysteresis. This strongly suggests that this folding is not in full equilibrium but trapped in a state that needs time to get resolved (entropic barrier?). Therefore, all conclusions based on the equilibrium assumption are limited. The FRET denaturation curves do not overlap either, suggesting that the buffer condition is not chosen such that the protein reaches full equilibrium and thus any information extracted to gather Gibbs free energy changes is questionable. For a discussion in literature I recommend Moon, ... Fleming (<https://www.ncbi.nlm.nih.gov/pmc/articles/PMC3193555/>) who explain why the full reversibility is important for equilibrium information. It would be very important for the author to justify how their equilibrium is a true equilibrium if it actually does not appear like an equilibrium? Isn't the definition of equilibrium that you reach the same state no matter from which direction you approach it?

I have a few additional comments:

- Figure legend 1A reads misleading with regard to: The two residues of the folding ... in dark grey and green respectively. Maybe easier to name the two residues again or include in previous sentence.
- Conceptually, I am puzzled by the mutation of the P298 at the end of the α -helix to a cysteine. My understanding is that it might be important for breaking the helix and thus limiting the secondary structure. Did the authors verify that the structure is conserved after mutation? Apparent stabilities seem shifted according to Fig. 1B.

- I am wondering about the conclusion why the additional barrier is an entropic barrier. Should the authors perform temperature dependent experiments to dissect enthalpies and entropic contributions for WT and DM-MBP? Also, if more dynamics are observed for some domains during folding in the DM-MBP, the transition barrier should be lower in that case and maybe ΔG reduced, allowing a population of multiple states much easier than with an increased ΔG .
- Just a comment: I am impressed by the high quality of the cross-correlation data for all three constructs.
- The three colour experiments are difficult- yet I also cannot fully follow the interpretation. The authors discuss the presence of an intermediate state. Do they mean the intermediate FRET population from interconversion dynamics? Would it make sense to remove the dynamic molecules from a state analysis using FRET-2CDE (Tomov 2012) based filters?

- The chaotropic salt GuanidiumCl is sometimes abbreviated GuHCL. Please change to GuHCl.

Reviewer #1 (Remarks to the Author):

The folding of large multi-domain proteins is understudied compared with the extensive work that has been done on small single domain proteins. For those multidomain proteins whose folding has been studied, proteins with a sequential "beads on a string" domain topology are the best characterized, while proteins with discontinuous, interdigitated, domains have received less attention.

This manuscript presents a very thorough study of the folding of a multidomain protein with a complex, discontinuous topology using multiple techniques. as such, it addresses an understudied aspect of protein folding.

I have a few questions.

In figure S3 (D, F, and H), the authors present refolding kinetics for the N and C terminal domains and the interface. It appears that all 3 regions fold on very similar timescales, and that the C terminal domain actually has a slightly shorter refolding $t_{1/2}$ than the N terminal domain. This seems inconsistent with the finding that the C terminal domain requires prior formation of the N terminal domain to fold. Do the authors have a proposed explanation for this?

First, the rates given in Figure S3 are approximations and the uncertainty, especially of the CTD measurements, are large. We believe that the differences in refolding times are due to the uncertainty in the measurements and think it would be an overinterpretation to discuss it as a potential difference in refolding rates. To make this clearer, we now plot the average and stdev of the three-independent measurements the figures Fig.S3 D,F,H rather than showing one representative curve. Thus, the values have changed slightly. To minimize confusion with regards to what we are measuring, we have relabeled the axis of the refolding kinetics.

On page 8 the authors write:

"It has been suggested earlier that DM-MBP refolding must overcome an entropic barrier (25) meaning that, in the absence of the hydrophobic core, a wide conformational search on a flat energy surface needs to be performed. If this were the case, we would expect refolding of DM-MBP at higher GuHCl concentrations to occur on longer timescales."

I am puzzled by the reasoning here. Denaturants are generally observed to slow the kinetics of protein refolding, regardless of the specific folding mechanism.

We thank the reviewer for pointing this out. The reviewer is correct that denaturants generally slow the refolding kinetics. What is unique in our case is that, at very low denaturant concentration (0.1-0.3 M GuHCl), there is a dramatic decrease in the refolding time by at least one order of magnitude. We have now reworded the text to be clearer:

The section now reads: " To investigate whether the hysteresis is due to not allowing the system enough time to reach equilibrium, we measured refolding of DM-MBP at 0.3 M GuHCl over three days. Indeed, the number of refolded proteins increased during the first 18 hours accounting mainly for the molecules shifting the FRET efficiency from 0.6 to 0.85 FRET efficiency (**Fig. S8**). Afterwards, no significant increase in the number of refolded molecules is observed, but there are changes in the dynamic properties of the unfolded intermediate. As no aggregation is visible after 48 h, we attribute this behavior to non-reversible misfolding of DM-MBP (**Fig. S8**). This suggests that hysteresis comes from the presence of an additional kinetic pathway leading to irreversible protein denaturation on the day timescale.

As has been suggested earlier, the slow refolding of DM-MBP is thought to be due the presence of an entropic barrier (25) meaning that, in the absence of the hydrophobic core, a wide conformational search on a flat energy landscape needs to be performed. If this were the case, we should expect a slight increase in denaturant at low concentration (0.1 to 0.3 M GuHCl) to have a significant impact on the refolding rate. Denaturants are known to slow refolding and GuHCl can replace interprotein hydrogen bonds, leading to more flexibility and competition for hydrogen bonding. However, for an enthalpic barrier, it is unlikely that the small change in denaturant from 0.1 to 0.3 M would lead to more than an order of magnitude difference in refolding time. To look for additional support for an entropic barrier to refolding, we also investigated the influence of the chemical-chaperone trimethylamine N-oxide (TMAO) on DM-MBP refolding. TMAO stabilizes the solvent shell around the protein thereby confining the configuration space available to the protein (31). DM-MBP refolding in the presence of TMAO is accelerated forming native-like structures already at 0.2 M GuHCl, where DM-MBP has a significant population of dynamic molecules in the absence of TMAO (**Fig. 2D**). Also, the rate of DM-MBP refolding only depends weakly on temperature (**Fig. S9, Table S1** and ref (32)). Taken together, these results suggest that the refolding of DM-MBP is limited by an entropic barrier."

Reviewer #2 (Remarks to the Author):

In this work, Agam et al. study the folding of a 2-domain protein DM-MBP, in isolation as well as in the presence of the GroEl/ES chaperone. Detailed studies of multidomain protein folding lag in the field, even though such proteins represent a majority of eukaryotic proteomes. In the current work, several powerful features of multiparameter single molecule fluorescence/FRET including 3-color experiments are leveraged nicely in the study. Interesting insights emerge, including presence of hysteresis, involvement of an entropic barrier and mechanistic understanding of the accelerated folding in the chaperone by reduction of the entropic barrier due to confinement. Especially exciting is the use of filtered FCS and 3-color smFRET to obtain direct information about the order of domain folding. The combination of new insights on a multidomain protein system and interesting application of various powerful features of multiparameter single molecule fluorescence will make this work of substantial interest. However, there are some issues that need to be addressed.

The authors discuss based on their data that the observed intermediate state is due to rapid

fluctuations between folded and unfolded protein states (e.g., page 8). Presumably that authors are saying that that this observed "intermediate state" is not a separate intermediate protein state, but rather a result of the measurement timescale being slower than the protein transition kinetics (the latter is discussed in the paper based on the data presented and is also a well-known kinetic phenomenon in the smFRET field including in the authors' work). Authors, please clarify the use of the "intermediate state" terminology/address as needed.

The reviewer is correct. What we call an "intermediate state" is indeed dynamic fluctuations between different states. We called it an intermediate state because it appears as such in the smFRET histograms, if you are unaware of the dynamic fluctuations. We have now rewritten the text to refer to this species as an intermediate population and are more explicit about what we mean with the intermediate population.

On page 18, it is unclear which data are showing the stretched conformation referred to in the text.

We thank the reviewer for pointing out this confusion. The stretched conformation is only in the presence of GroEL. It is observable in Figure 5B (the major population with a donor lifetime around 3.8 ns). We now clarify this better in the text: "...two populations are observable, an intermediate population which falls on a red dynamic FRET line and an extended or a stretched conformation on the static FRET line with a donor fluorescence lifetime of ~3.8 ns (**Fig. 5B**) as we have also observed previously (24)."

The authors discuss that the work shows that "the unfolding/refolding hysteresis of DM-MBP occurs due to rapid dynamics between extended and compact conformations on the micro- to millisecond timescale". Authors please clarify how such rapid fluctuations may be giving rise to hysteresis on an orders of magnitude slower timescale.

We thank the reviewer for point out this miswording. Indeed, it is not the rapid dynamics that lead to the hysteresis. What we meant to say is that, under the conditions where hysteresis is present, we observe the rapid dynamics between extended and compact conformations on the micro- to millisecond timescale.

We have changed the text to now read: "Our study shows that the previously known unfolding/refolding hysteresis of DM-MBP occurs under conditions where the protein undergoes rapid dynamics between extended and compact conformations on the micro- to millisecond timescale rather than aggregation effects (46) (**Fig S1**) or a kinetically-trapped state (**Figs. 2-3**) (25)."

In the labeling for the 3 color smFRET experiments, do the authors have data supporting site specific labeling? In particular, while the strategy of blocking Cys 298 with maltose is neat, if the labeling efficiency is not high, the subsequent labeling step with Atto 565 maleimide will react with the remaining unlabeled Cys 175 in addition to Cys 298. Such a mixture might contribute to/complicate the smFRET data, in particular to the BG component. Authors please address.

As the reviewer correctly points out, it is important to have specific labeling for the three-color FRET experiments. However, the important issue for our experiment is how well Cys 298 is blocked when labeling position Cys 175, which is demonstrated in Figure S12, rather than the labeling efficiency of Cys298. Due to the high maltose concentration used, it is unlikely that Alexa647 reacts with Cys298 rather than Cys 175 within the limited labeling time of 1 hr and only 2-fold excess of dye used. The average labeling efficiency of Cys298 is 40 %. However, even though the labeling efficiency is not optimal, unlabeled Cys298 is not a problem when labeling Cys 175 with Atto 565. If Cys298 is unlabeled when starting the labeling reaction with Atto 565, the protein will not contain the Alexa647 label. Hence, even if Cys298 is labeled with Atto 565, the molecule will lack all three fluorophores and will not be included in further analyses.

We now state this explicitly in the text in the materials and methods section: "The cysteine-maleimide reaction was performed by the addition of 2-fold molar excess of Alexa647-maleimide and allowed to react for 1 hr at RT. Under these conditions, the overall labeling efficiency of position K175C is 40%. However, more important than the labeling efficiency at this point is minimizing the possibility of mis-labeling the maltose blocked cysteine at position 298 with Alexa 647 (**Fig. S12**). Molecules that lack all three dyes are not included in further three-color analyses, whereas molecules with the dyes labeled unspecifically would complicate the analysis."

On page 5, a nice FCCS control is used to rule out aggregation. The authors may choose to mention the experiment briefly in the main text for readers' benefit.

We now mention the FCCS experiment in the main text as "We confirmed that the delayed refolding is not caused by aggregation using fluorescence cross-correlation spectroscopy (FCCS) experiments during refolding of DM-MBP. Equal amounts of a single-cysteine mutant of DM-MBP (A52C) labeled with either Atto532 or Alexa647 were denatured in 3 M GuHCl at a concentration of 500 nM. The samples were then mixed and diluted 25-fold to a final total protein concentration of 40 nM in 0.1 M GuHCl. The final denaturant concentration of 0.1 M GuHCl was chosen as refolding to the native state still occurs under these conditions. There was no detectable cross-correlation amplitude during 60 minutes of refolding, suggesting that the delayed refolding of DM-MBP is caused by a slowly refolding population (**Supplementary Text, Fig. S1C**)."

For Figure 1B discussion of hysteresis on page 5, it is worth noting the time delay after making the sample for the experiment.

We have now added this information in the text:

"Sample were allowed to equilibrate under unfolding/refolding conditions for 20 hours before measuring."

and later

"We measured the unfolding/refolding trajectory of the NTD conformation at different GuHCl concentrations after allowing the sample to equilibrate for 2 hours (**Fig. 1B, C**)."

Presumably the N-C mutant combines FRET information about the interface and folding of NTD/CTD. Authors, please briefly comment on this point.

We are probing the distance between position 52 and 175. When the two lobes of the NTD and CTD come together, this interaction should be stabilized. We assume that a well-defined FRET signal for the N-C interface requires a good, overall folding of both domains. However, as the three-color experiment show, the situation is more complicated. The N-C interface can form a compact structure even when the CTD is not yet folded. This implies that the lower alpha helices of the CTD can fold and interact with the NTD interface, even when the rest of the CTD, which is dependent on the discontinuous region of the protein, has not yet folded. We now mention this in the text: "In addition, the N-C interface can form when the α -helices harboring the 175 labeling position in the CTD is stabilized together with the folded NTD.

Reviewer #3 (Remarks to the Author):

In this manuscript, Agam and co-workers studied the folding of DM-MBP, an MBP variant with reduced folding rates, using smFRET spectroscopy. In their study they monitor folding of the C-terminal domain, the N-terminal domain and the overall protein using FRET pairs on different locations. They further perform three colour FRET experiments to simultaneously monitor DM-MBP folding and also monitor folding in presence of GroEL/ES. They find that the major folding event is limited by an entropic barrier, which is still present in GroEL/ES but reduced.

This manuscript is from the smFRET side rigorously executed. Fluorophore positions are carefully chosen and the data is of high quality and analysed to recently defined standards in the field. The biological model system DM-MBP is very well established and very well studied. This is a point, where I was struggling most, because it has not become clear to me where the real novelty of the study is. These mutations are known for a long time and the kinetic trapping as well during the folding from ensemble experiments, therefore I see here limited new insights - as also indicated by the. discussion section of the manuscript.

We thank the reviewer for her/his praise for the quality of the data and analyses performed. Biologically speaking, we learn that the order of domain folding is preserved in this discontinuous two-domain protein, even though folding of the NTD is slowed down by over an order of magnitude. Also, we discovered that an intermediate population was actually dynamic, fluctuating between unfolded and compact conformations. This indicates that the protein is not stuck in a kinetically trapped state. More generally, we demonstrate the importance of dynamics fluctuations during refolding processes. This dynamic population was also observed when bound to a chaperone protein (i.e. attached to GroEL and when encapsulated within the cavity of GroEL/GroES). Hence, dynamics is also important in the context of chaperone function. Using three-color FRET, we could directly confirm what had been speculated in the previous studies. In addition, we also demonstrate how three-color FRET measurements can be utilized to gain insights into the very complex and highly dynamic process of protein folding.

When demonstrating a new approach, it is important to demonstrate the capability and potential of the method on well known systems before applying it to cases that are less well understood. The approach that we used here can be generalized to investigating coordinated motions of proteins and thereby gaining mechanistic insights into allosteric effects, to investigate protein-chaperone interactions, which is related to diseases involving protein aggregation and also provides relevant information to researchers performing protein engineering and design. We have now extended the discussion section to bring out these points more.

One of the major challenges in the model system is in my eyes the reversibility. The authors show already in Fig 1B that even though performed presumably at equilibrium, the unfolding and refolding TRP curves do not overlap but show a huge hysteresis. This strongly suggests that this folding is not in full equilibrium but trapped in a state that needs time to get resolved (entropic barrier?). Therefore, all conclusions based on the equilibrium assumption are limited. The FRET denaturation curves do not overlap either, suggesting that the buffer condition is not chosen such that the protein reaches full equilibrium and thus any information extracted to gather Gibbs free energy changes is questionable. For a discussion in literature I recommend Moon, ... Fleming (<https://www.ncbi.nlm.nih.gov/pmc/articles/PMC3193555/>) who explain why the full reversibility is important for equilibrium information. It would be very important for the author to justify how their equilibrium is a true equilibrium if it actually does not appear like an equilibrium? Isn't the definition of equilibrium that you reach the same state no matter from which direction you approach it?

The reviewer is correct that, due to the hysteresis, we are not fully in equilibrium. The thermodynamic ground state is only reached upon the final folding of the protein. What we are measuring is a pseudo equilibrium of the dynamic fluctuations between the unfolded and compacted (but not yet fully folded) states. Hence, we have now added an additional state (the compact state) to our model to address this (see Figure 3D). The dynamics between the unfolded and compact state is very fast with respect to the folding time and the measurement time. Hence, we can treat it as a pseudo equilibrium. We referred to the measurements as "equilibrium" measurements to distinguish them from the kinetic measurements. In these experiments, we allowed the system to reach a quasi steady-state before starting the measurement. We have also renamed the rates obtained from PDA as apparent unfolding and apparent refolding rates representing the pseudo-equilibrium between the unfolded and compact states. The pseudo equilibrium smFRET experiments are performed after 1-2 hrs of incubation and no changes in the smFRET histograms are observed during the experiments. Similarly, samples were equilibrated for 20 hrs to reach the quasi-steady state before measuring the Trp fluorescence.

In the interesting work of Moon et al, the authors were able to get rid of the hysteresis by avoiding aggregation and by using the appropriate pH so that the membrane protein has no problem being inserted into the membrane. In our case, neither protein aggregation or pH is causing the hysteresis. We note that the hysteresis in DM-MBP can be eliminated by adding disulfide bonds in the protein to limit the conformational entropy (Chakraborty, 2010).

Unfortunately, due to the added cysteines used in this mutant, we can no longer label them for smFRET measurements.

The fact that the smFRET denaturation curve does not entirely overlap with the tryptophan measurements is expected as the two methods are sensitive to different phenomena. We discuss this in more detail in the related comment below.

To approximate the free energy of refolding, we calculate the free energy based on the refolding rate and relaxation times at 0.1 M denaturant. For sake of calculation, we assume a simple model and approximate the protein reconfiguration time for folding between the compact and folded state from the relaxation times for the fluctuations between unfolding and the compact state. These are much faster than folding and are in a clear pseudo equilibrium. However, as we are not able to distinguish between the compact and folded state, trapped states may also be involved in the last step of folding. We now discuss this in more detail in the discussion: "Although, we cannot rule out the existence of a trapped, compact state during refolding, we assume the reversibility of the fluctuations between the unfolding and compact state and use a simple folding model with an entropic barrier to estimate the free energy. The reconfiguration times during refolding were approximated from the relaxation times obtained using dynamic PDA."

Chakraborty K, Chatila M, Sinha J, Shi Q, Poschner BC, Sikor M, Jiang G, Lamb DC, Hartl FU, and Hayer-Hartl M. Chaperonin-Catalyzed Rescue of Kinetically Trapped States in Protein Folding. *Cell* 2010 142: 112-122.

I have a few additional comments:

- Figure legend 1A reads misleading with regard to: The two residues of the folding ... in dark grey and green respectively. Maybe easier to name the two residues again or include in previous sentence.

We have reworded the sentence to read: " The two residues of the folding mutations (V8G and Y283D, highlighted in dark grey) as well as the three labeling positions A52, K175, P298 (highlighted in green) for coupling the fluorescent dyes are indicated via a space filling model. "

- Conceptually, I am puzzled by the mutation of the P298 at the end of the α -helix to a cysteine. My understanding is that it might be important for breaking the helix and thus limiting the secondary structure. Did the authors verify that the structure is conserved after mutation? Apparent stabilities seem shifted according to Fig. 1B.

Position P298C has been used in many previous studies and the mutation has been shown to not influence the protein structure or maltose binding (Sharma, 2008; Chakraborty, 2010). The shift shown in Figure 1B is due to the fact that you are measuring the unfolding/refolding using similar, but different methods that are sensitive to different things. For tryptophan fluorescence, you are measuring the average quenching of the tryptophanes due to water accessibility. Hence, it is sensitive to whether water can penetrate to the locations of the tryptophanes. FRET is measuring the actual distance between the two fluorophores. Hence,

while both are a measure for the conformational structure of the protein, it is not surprising that there is a shift between the two methods. We now mention this explicitly in the text: "We note that there are differences between the unfolding / refolding curves measured via tryptophan fluorescence versus FRET efficiency. While both methods are used to investigate protein unfolding/refolding, tryptophan fluorescence measures the solvent accessibility due to quenching by water whereas FRET measures the actual distance between the fluorophores. Hence, some differences are expected."

Sharma S, Chakraborty K, Muller BK, Astola N, Tang YC, Lamb DC, Hayer-Hartl M, and Hartl FU. Monitoring protein conformation along the pathway of chaperonin-assisted folding. *Cell* 2008 **133**: 142-153.

Chakraborty K, Chatila M, Sinha J, Shi Q, Poschner BC, Sikor M, Jiang G, Lamb DC, Hartl FU, and Hayer-Hartl M. Chaperonin-Catalyzed Rescue of Kinetically Trapped States in Protein Folding. *Cell* 2010 142: 112-122.

- I am wondering about the conclusion why the additional barrier is an entropic barrier. Should the authors perform temperature dependent experiments to dissect enthalpies and entropic contributions for WT and DM-MBP? Also, if more dynamics are observed for some domains during folding in the DM-MBP, the transition barrier should be lower in that case and maybe ΔG reduced, allowing a population of multiple states much easier than with an increased ΔG .

The original suggestion that the slow-folding of DM-MBP is due to entropy comes from our previous work (Chakraborty et al, *Cell* 2010). Restricting the conformational space available to DM-MBP, either by the presence of a chaperone, GroEL/GroES or by the addition of disulfide bonds, accelerated refolding of DM-MBP. Experiments in the presence of the chemical chaperone, TMAO, also shifted the equilibrium towards the folded state. We have measured the temperature dependence of the folding rate for DM-MBP using smFRET (Supplementary Figure S9). We show that, for temperatures above 25 °C, there is no measurable change in the refolding rate, consistent with this hypothesis. Due to the fast refolding of WT MBP, smFRET experiments of the refolding process is not currently feasible in our laboratory. Hence, this is the model we favor. We have now reworded the text to be more clear about the different processes and states we are measuring.

The ΔG we are referring to is the free energy of folding (assuming a simplified folding model). The dynamic population is the transition between the unfolded and compact state. The final folding step is between the compact and folded state. We have now updated the diagram in Figure 3C (now Figure 3D) to show the different states.

- Just a comment: I am impressed by the high quality of the cross-correlation data for all three constructs.

- The three colour experiments are difficult- yet I also cannot fully follow the interpretation. The authors discuss the presence of an intermediate state. Do they mean the intermediate FRET

population from interconversion dynamics? Would it make sense to remove the dynamic molecules from a state analysis using FRET-2CDE (Tomov 2012) based filters?

We apologize for the confusion. What we meant is an intermediate FRET population, which is dynamically converting between an unfolded and a compact family of structures. We have reworded the text to now refer to the intermediate state as an intermediate population. We experimented with using the FRET-2CDE filter to remove the dynamic populations, but it appears that the dynamics are too fast.

- The chaotropic salt GuanidiumCl is sometimes abbreviated GuHCL. Please change to GuHCl.

We have corrected the typo in the text.

Reviewers' Comments:

Reviewer #1:

Remarks to the Author:

This revised manuscript addresses my previous concerns.

Reviewer #2:

Remarks to the Author:

The authors have done a good job of addressing my previous round of comments. This combination of new insights on a multidomain protein system using multiparameter single molecule fluorescence will be of substantial interest to the scientific community.

Reviewer #3:

Remarks to the Author:

Agam and co-workers have presented a revised manuscript on the study of folding of the multi-domain protein maltose binding protein and a weakened variant of it. They use smFRET as a two-color and three-color approach in buffer and in complex with the chaperone GroE.

I appreciate the revision of the manuscript and the careful answers to my points but also the points by the other reviewers. With regard to my raised question about equilibrium, I value the careful evaluation and the introduction of a new compact state (of yet unknown nature) to resolve this conundrum. While this indeed provides a route out of the observed discrepancy, I would appreciate two more considerations by the authors:

- What happens in the chaperone? Is there the additional compact state also present? Figure 5E does suggest otherwise. Is this the effect of the chaperone? Or is this compact state also observed there?

- I think the compact state might indeed be an important state that is occurring during the folding pathway and the authors study important kinetics to this state and dynamics around this state. The manuscript, especially the abstract and introduction, currently does not mention this additional compact state. I would appreciate if the authors mention this at least at the end of their introduction to make the reader aware that not the fully folded state is probed suggesting an even more complex topology of folding of MBP.

REVIEWERS' COMMENTS

Reviewer #1 (Remarks to the Author):

This revised manuscript addresses my previous concerns.

Reviewer #2 (Remarks to the Author):

The authors have done a good job of addressing my previous round of comments. This combination of new insights on a multidomain protein system using multiparameter single molecule fluorescence will be of substantial interest to the scientific community.

Reviewer #3 (Remarks to the Author):

Agam and co-workers have presented a revised manuscript on the study of folding of the multi-domain protein maltose binding protein and a weakened variant of it. They use smFRET as a two-color and three-color approach in buffer and in complex with the chaperone GroE.

I appreciate the revision of the manuscript and the careful answers to my points but also the points by the other reviewers. With regard to my raised question about equilibrium, I value the careful evaluation and the introduction of a new compact state (of yet unknown nature) to resolve this conundrum. While this indeed provides a route out of the observed discrepancy, I would appreciate two more considerations by the authors:

- What happens in the chaperone? Is there the additional compact state also present? Figure 5E does suggest otherwise. Is this the effect of the chaperone? Or is this compact state also observed there?

We thank the reviewer for her/his input and appreciation of our responses. The compact states contain similar separations between the donor and acceptor fluorophores and then similar FRET efficiencies to the folded state, but the protein has not locked into the folded conformation. Hence, we need to use the existence of dynamics to distinguish between the compact states and the folded state (for measurements at GuHCl concentrations where hysteresis is visible). As we have dynamics within the chaperone, we expect that at least a sub-set of the same family of compact states exists within the chaperone. In Figure 5E, the conformational search is represented by the flat surface of the folding funnel. The flat region is

present for both DM-MBP alone and in the chaperone, although with a restricted area as there is less conformational freedom within the chaperone. We now mention this point when discussing the folding of DM-MBP within the chaperone and have corrected Figure 5E to bring out this point more clearly.

- I think the compact state might indeed be an important state that is occurring during the folding pathway and the authors study important kinetics to this state and dynamics around this state. The manuscript, especially the abstract and introduction, currently does not mention this additional compact state. I would appreciate if the authors mention this at least at the end of their introduction to make the reader aware that not the fully folded state is probed suggesting an even more complex topology of folding of MBP.

We have now edited the abstract and the introduction to mention the compact state, as the reviewer suggested.